# Kif2 localizes to a subdomain of cortical endoplasmic reticulum that drives asymmetric spindle position

Vlad Costache[1], Celine Hebras[1], Gerard Pruliere[1], Lydia Besnardeau[1], Margaux Failla[1], Richard R. Copley[1], David Burgess[2], Janet Chenevert[1] & Alex McDougall[1]

Asymmetric positioning of the mitotic spindle is a fundamental process responsible for creating sibling cell size asymmetry; however, how the cortex causes the depolymerization of astral microtubules during asymmetric spindle positioning has remained elusive. Early ascidian embryos possess a large cortical subdomain of endoplasmic reticulum (ER) that causes asymmetric spindle positioning driving unequal cell division. Here we show that the microtubule depolymerase Kif2 localizes to this subdomain of cortical ER. Rapid live-cell imaging reveals that microtubules are less abundant in the subdomain of cortical ER. Inhibition of Kif2 function prevents the development of mitotic aster asymmetry and spindle pole movement towards the subdomain of cortical ER, whereas locally increasing microtubule depolymerization causes exaggerated asymmetric spindle positioning. This study shows that the microtubule depolymerase Kif2 is localized to a cortical subdomain of endoplasmic reticulum that is involved in asymmetric spindle positioning during unequal cell division.

[1] Sorbonne Universités, UPMC Univ Paris 06, CNRS, Laboratoire de Biologie du Développement de Villefranche-sur-mer (LBDV), Observatoire Océanologique, Villefranche sur-mer 06230, France. [2] Boston College, Biology Department, 528 Higgins Hall, 140 Commonwealth Ave, Chestnut Hill, MA 0246, USA. Correspondence and requests for materials should be addressed to J.C. (email: chenevert@obs-vlfr.fr) or to A.M. (email: dougall@obs-vlfr.fr)

Control of microtubule dynamics at the cell cortex is important for a myriad of processes including spindle positioning at the cell center during cell division[1–3], asymmetric spindle positioning during unequal cell division (UCD) in embryos[4, 5], and for axonal pruning during nervous system development in mammals[6, 7]. Microtubule dynamics have been intensively studied during asymmetric cell division (ACD) which is sometimes coupled with UCD creating one large and one small daughter cell that have different fates, as in *Drosophila*

neuroblasts[8], sea urchin micromeres[9], and *C. elegans* 1-cell zygotes[10]. Such UCD has two components: cortical pulling forces acting on astral microtubule plus ends[11] and depolymerization of microtubule plus ends as they encounter the cortex[12, 13], which together create unbalanced forces to position the mitotic spindle asymmetrically. These two processes acting together overcome the forces that cause mitotic spindles to move to the center of the cell, a process that senses and integrates force over the length of microtubules[14]. However, one key piece missing from this model

**Fig. 1** Microtubules and centrosome-attracting body (CAB) during UCD. **a** Aster behavior during unequal cell division. Schematics showing embryos from the 8–64-cell stage together with bright-field images of embryos with the 2 blastomeres that undergo unequal cell division highlighted *green*. Right: selected confocal planes from live 4D imaging experiments showing microtubule organization in the pairs of blastomeres that undergo UCD at the 8, 16, and 32-cell stages (corresponding to the schematics). All microtubules were labeled in live embryos with Ensconsin::3GFP. *Scale bars* = 30 μm. See Supplementary Movies 2, 3, 4, and 5. **b** CAB visualization. *Left image*: 3D rendering of several confocal imaging planes reveals the CAB in the two *bottom* blastomeres (B4.1 pair) at the 8-cell stage in a live embryo (*red arrows*). CAB visualized with PH::Tom (*red*) and the deeper cytoplasm with the microtubule-binding protein EB3::GFP (*green*). See Supplementary Movie 6. *Center image*: live confocal imaging of the cortical endoplasmic reticulum in the CAB (*labeled red*) with a marker for endoplasmic reticulum (DiIC16) and the mitochondria (*labeled green*) with Mitotracker. The cER present in the CAB (also see deeper ER accumulated on the mitotic spindles) is attached to a specialized apical domain. *Right image*: confocal image of a fixed 16-cell stage embryo stained with antibodies to aPKC (*green*) and the deeper and surrounding mitochondria with NN18 (*red*). Chromosomes stained with DAPI (*blue*). Note the dark unlabeled zone between the CAB surface and the mitochondria which is filled with cER (see dotted line at one interface for clarity). *Scale bars* = 30 μm

of UCD is the identity of the protein(s) that cause astral microtubule plus end depolymerization at the cortex, which is important not only for UCD, but also for the mitotic spindle centering mechanism based on astral microtubule length that operates during symmetric cell division.

Microtubule plus ends can be induced to depolymerize via different mechanisms. In vitro experiments indicate that dynein can cause catastrophe of microtubule plus ends[15], raising the possibility that in intact cells dynein couples pulling with depolymerization. A different mechanism regulates microtubule plus end depolymerization in developing mammalian neurites which is dependent on the cortically localized microtubule depolymerase Kif2A[6]. Kif2A is a member of the kinesin-13 family of microtubule depolymerases[16] which includes MCAK/Kif2C that causes microtubule plus end depolymerization at kinetochores during anaphase[17]. However, in cells that divide unequally it is still not known what causes astral microtubule plus end depolymerization at the cortex. In C. elegans one protein has been described (EFA-6) which limits cortical microtubule growth, however the knockdown of EFA-6 does not prevent UCD[18].

C. elegans embryos have provided a wealth of knowledge about the cortical pulling forces that act upon astral microtubules during UCD. For example, following fertilization and symmetry breaking in C. elegans, the Par polarity complexes are partitioned to distinct cortical subdomains[19]. Anterior Par3/Par6/aPKC (PKC-3) phosphorylates LIN-5 (NuMA) at the anterior cortex inhibiting the cortical anterior spindle pulling forces[20],while NuMAs binding partner GPR-1/2 (Pins/LGN) becomes enriched at the posterior cortex during mitosis[21–23]. The dynein light chain protein DYRB-1 coupled to GFP has been demonstrated to co-immunoprecipitate with endogenous LIN-5 and GPR-1/2 in C. elegans embryos thus suggesting that DYRB-1 may provide a physical link between the endogenous dynein/dynactin complex and either LIN-5 or GPR-1/2[24]. However, this interaction has not been shown to be limited to the posterior cortex. During metaphase and anaphase the mitotic spindle is pulled towards the posterior cortex causing UCD[11]. Late in mitosis the posterior centrosome has changed from a spherical shape to a flattened and disk-shaped structure[25]. Symmetric cell divisions in somatic cells also depend upon cortical dynein to center the mitotic spindle[2, 26, 27]. In C. elegans, in addition to dynein pulling forces, it has been shown that astral microtubule depolymerization also plays a role in posterior spindle displacement. For example, pulling forces are lacking when microtubules are stabilized by taxol, while in embryos carrying a temperature-sensitive mutation in a β-tubulin gene the posterior displacement distance of the spindle is enhanced[28]. Based on these and other data a dual force-generation mechanism has been proposed that relies on microtubule pulling forces (dynein-dependent) combined with microtubule depolymerization[13, 28]. Thus in C. elegans, although cortical microtubule depolymerization is thought to be part of the mechanism for posterior spindle displacement, the mechanism regulating cortical astral microtubule depolymerization is not known[19].

Many embryos provide more extreme examples of UCD whereby the two asters of the mitotic spindle become highly asymmetric in size and shape with the smaller of the two asters being inherited by the smaller of the two daughter cells. Similar to the flattened posterior centrosome in C. elegans one-cell embryos[25], such mitotic aster asymmetry has also been observed during UCD in spiralian[29, 30], echinoderm[31, 32], and ascidian[33, 34] embryos. Mitotic aster asymmetry commonly occurs at the third cleavage in spiralian embryos with the smaller aster associating with an animal cortical domain at the 4-cell stage leading to UCD[29]. A similar phenomenon has been documented for sea urchin embryos where one aster in each future micromere

associates with a cortical domain situated at the vegetal pole of the 8-cell stage embryo creating aster asymmetry during UCD[31]. The clearest example of a cortical domain that causes aster asymmetry and UCD comes from ascidian embryos. Here a large cortical structure termed the centrosome-attracting body (CAB) causes three successive rounds of UCD accompanied by the CAB-proximal aster becoming smaller[33–35]. Although the Par complex (aPKC, Par3, and Par6) is localized to the CAB[36], we do not currently know how the CAB affects microtubule dynamics leading to aster asymmetry. Due the large dimensions of the CAB (circa 20 μm at the 8-cell stage[37] and this article) we wondered whether it would be possible to identify proteins involved in regulating microtubule dynamics at the cortex via live-cell imaging during UCD in the early ascidian embryo.

We have developed the optically transparent ascidian species Phallusia mammillata as a system to perform live-cell imaging to study microtubule dynamics during UCD[34, 38]. By analyzing microtubule dynamics at the cortex we discovered that a microtubule depolymerase (Kif2) localizes to the cortical CAB in a cell-cycle-dependent manner. Through live-cell imaging of Kif2::Venus/mCherry/Tomato and immunofluorescence we demonstrate that exogenous and endogenous Kif2 localizes to the cortical CAB. In particular, Kif2 accumulates on a subdomain of cortical endoplasmic reticulum (cER) concentrated at the CAB during interphase and leaves the CAB cER during mitosis when CAB-proximal microtubules become short. In addition, we show that microtubules are less abundant in the cortical CAB during interphase. Finally, we found that inhibiting endogenous Kif2 protein function prevents the establishment of mitotic aster asymmetry, and conversely that increasing depolymerization of microtubules near the subdomain of cER at the CAB causes exaggerated asymmetric spindle positioning.

## Results

**Asymmetric spindle positioning and the CAB.** The early embryo of the European ascidian Phallusia mammillata is favorable for live-cell imaging and functional studies because its cells are transparent (see Supplementary Movie 1) and readily translate exogenous messenger RNAs (mRNAs) such as those encoding GFP fusions and dominant negative constructs[38]. Ascidian embryos display three successive rounds of UCD that depend upon the CAB[33–35]. During these three rounds of UCD, one pole of the mitotic spindle is attracted to the CAB (Fig. 1a). As UCD ensues, astral microtubules emanating from the centrosome nearest the CAB and midline become shorter than those microtubules originating from the more distant centrosome (Fig. 1a). A smaller and flattened aster thus forms nearest the CAB and midline during each round of UCD (Fig. 1 and Supplementary Movies 2, 3, and 4 with a 3D rendering of an 8-cell stage embryo shown in Supplementary Movie 5). In Phallusia embryos, both centrosomes appeared similar for γ-tubulin staining (Supplementary Fig. 1) indicating that aster size is not a function of γ-tubulin loss from one centrosome as has been observed in leech zygotes[39].

The CAB is a multilayer structure and can be visualized in several ways[36, 40]. Because the CAB creates a protrusion it can be visualized with plasma membrane markers such as PH::Tom (Fig. 1b and Supplementary Movie 6). Also, because the CAB is rich in cER and excludes mitochondria it can be visualized by specific lipophilic dyes that label either the mitochondria or cER in the CAB (Fig. 1b). Finally, antibodies to aPKC label the cortical surface of the CAB but do not label the cER in the CAB which appears as a dark zone surrounded by mitochondria labeled with anti-mitochondrial antibody-NN18 (Fig. 1b).

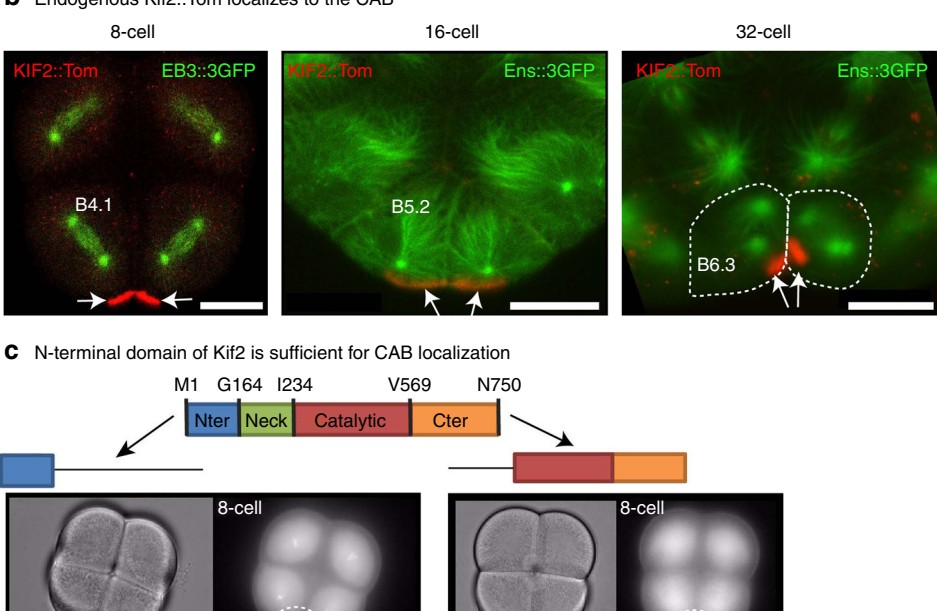

**Fig. 2** Kif2 protein localizes to the CAB. **a** Endogenous Kif2 in the CAB. Confocal images of two fixed 16-cell stage embryos stained with anti-Kif2 from two species of ascidian: *Phallusia mammillata* and *Ciona intestinalis* showing the accumulation of endogenous Kif2 protein (*green*) in the blastomeres (B5.2 pair) containing the CAB (*boxed*). DNA stained with DAPI (*blue*), microtubules with anti-Tubulin (*red*). $n = 94$ embryos. *Scale bars* = 30 μm. Images are representative of all embryos. **b** Exogenous Kif2::Tomato localizes to the CAB. Live *Phallusia* embryos expressing Kif2::Tom (*red*) and the microtubule markers EB3::GFP (*green*, 8-cell stage prometaphase, also see Supplementary Movie 7 where bright-field data has been included) or Ens::3GFP (*green*, 16-cell stage and 32-cell stage interphase) showing the localization pattern of Kif2::Tom to the CAB (*red*). *Arrows* indicate CABs. Scale bar = 30 μm (8-cell), 30 μm (16-cell), and 20 μm (32-cell). $n = 30$. Images are representative of all embryos. **c** N-terminal domain of Kif2 is sufficient for CAB localization. Schematic of Kif2 protein showing N-terminal domain, neck region, catalytic domain, and C-terminal domain. In order to determine which part of the protein was required for CAB localization different truncated versions of Kif2 were fused to Venus and their localization to the CAB followed by epifluorescence. *Left*: bright-field and epifluorescence images of 8-cell stage embryo expressing N-ter Kif2::Venus (truncated to amino acids 1-72). *Right*: bright-field and epifluorescence images of 8-cell stage embryo expressing truncated N-ter Kif2::Venus (amino acids 239-726). CAB region is indicated by oval. *Scale bar* = 30 μm. $n = 16$. Images are representative of all embryos

**Characterization of Kif2 in ascidian embryos**. In order to understand how the CAB may be involved in creating aster asymmetry we searched for CAB-resident proteins by screening likely candidates by either probing with antibody for immuno-fluorescence or localization of expressed tagged proteins (GFP, Venus, mCherry, or Tomato). We identified a number of proteins including Kif2, a member of the kinesin-13 family of proteins that instead of possessing motor activity displays microtubule depolymerization activity[41]. Vertebrates contain three members of the Kif2/kinesin-13 family: Kif2a, Kif2b, and MCAK (Kif2c)[42]. There is only one member of the Kif2 family in the ascidian (*P. mammillata*: PmKif2: unique gene ID: phmamm.g00002556 and *C. intestinalis*: CiKif2: unique gene ID: Ciinte.g00008837) and other non-vertebrate deuterostomes (Supplementary Fig. 2), suggesting that the vertebrate family of proteins evolved from a non-

vertebrate deuterostome Kif2a/b/c (henceforth Kif2). In two species of ascidian (*Phallusia mammillata* and *Ciona intestinalis*), Kif2 is a CAB-resident protein (Fig. 2a and Supplementary Fig. 3). Ascidian Kif2 also localizes to centrosomes and spindle micro-tubules (like Kif2a/Kif2b) although CAB localization was strongest (Supplementary Fig. 3). Overexpressing Kif2 to high levels shortened microtubules consistent with it acting as a microtubule depolymerase (Supplementary Fig. 4 and Supplementary Movie 14). By Western blot Kif2 antibody recognizes both endogenous Kif2 and injected Kif2 coupled to GFP (Supplementary Fig. 5). Finally, exogenous Kif2::Tom localized to the CAB during UCD at the 8-cell, 16-cell, and 32-cell stages in *Phallusia* embryos (Fig. 2b).

Next we wished to pinpoint which domain of Kif2 protein was required for CAB localization. For this we injected mRNAs

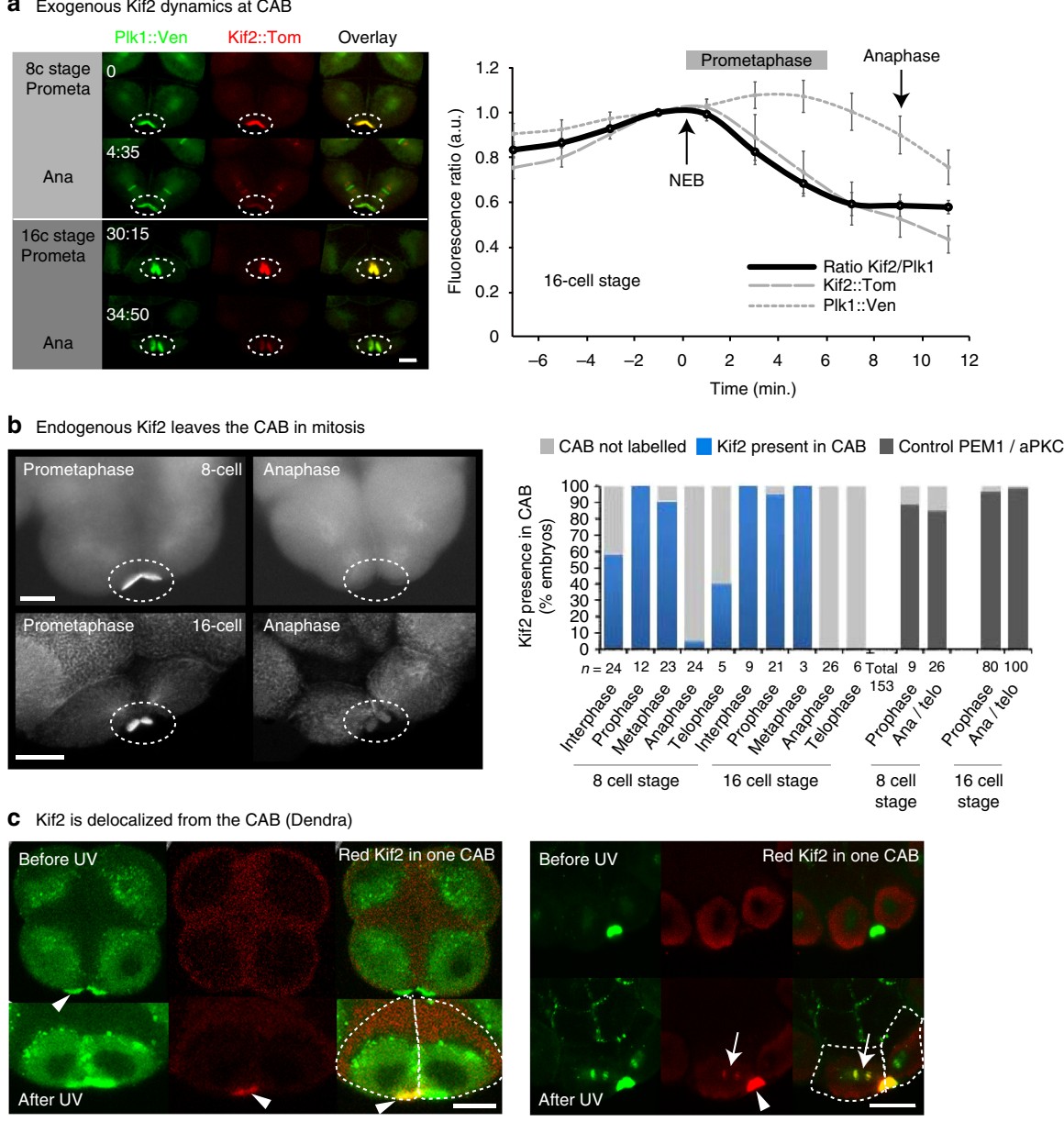

**Fig. 3** Kif2 dynamics at the CAB. **a** Exogenous Kif2 dynamics at CAB. Unfertilized eggs were microinjected with mRNA encoding Kif2::Tom and Plk1::Venus which also labels the CAB. Confocal images from a time-lapse series showing Plk1::Ven, Kif2::Tom localization at prometaphase and anaphase at the 8 and 16-cell stages. Time in min. Oval indicates CAB region. n = 6. Scale bar = 20 µm. Quantification of Plk1::Ven fluorescence versus Kif2::Tom fluorescence in the CAB. Ratiometric signal normalized to time of nuclear envelope breakdown (NEB). Plk1::Ven (*gray dotted line*), Kif2::Tom (*gray dashed line*) and ratio of Kif2::/Plk1 fluorescence (*solid black line*) are shown. n = 6. Kif2 fluorescence begins to decrease $42 \pm 17$ s (mean ± s.e.m.) following NEB (n = 9). See Supplementary Movie 8. **b** Endogenous Kif2 leaves the CAB in mitosis. Fixed 8-cell and 16-cell stage embryos probed with anti-Kif2 during metaphase (*left*) and anaphase (*right*); CAB region outlined by oval. *Scale bars* = 20 µm. Quantification of the immunofluorescence data at the 8–16 and 16–32-cell stages (cell cycle stages are indicated). Kif2 in the CAB was scored as present or absent regardless of the amount. The 4 bars at *right* are controls showing that two other CAB-resident proteins (PEM1 and aPKC) remain in the CAB at Anaphase/Telophase. n is indicated below each column. **c** Kif2 is delocalized from the CAB. Embryos at the 8-cell stage containing Kif2::Dendra. *Left*: *upper row* of images show Kif2::Dendra localization at the CAB (before UV photoconversion, *green*). Note absence of red Kif2 at the CAB. A small region of interest within the left CAB (*arrowhead*) was illuminated with UV light to cause photoconversion of the green Kif2::Dendra into red Kif2::Dendra (see *highlighted lower row* of images). *Left*: *lower row* of images following a brief UV illumination in a pre-defined region of interest centered in only one CAB, the photo-converted red Kif2 is created in one CAB (*arrowhead* indicates CAB-containing red Kif2). *Right*: *upper row* of images show localization of Kif2::dendra before photoconversion. *Right*: *lower row* of images show that some red Kif2::Dendra in CAB to the *left* (*strong red signal, arrowhead*) diffused away from the CAB to label the nearby chromosomes (*arrows*). The chromosomes in the adjacent blastomere to the right appear only *green* and are not labeled with the red Kif2::Dendra since the UV laser was focused on the *left* blastomere CAB (thereby creating red Kif2::Dendra only in the *left* CAB). *Scale bars* = 30 µm. n = 6

encoding fusions between Venus tag and various portions of Kif2 and evaluated CAB localization of each construct. Our results show that the N-terminal 72 amino acids of Kif2 was capable of targeting Kif2 to the CAB, and conversely removing the N-terminal domain abolished CAB localization of the Venus-tagged constructs (Fig. 2c). The N-terminal domain of MCAK is involved in subcellular targeting[43] which is consistent with our findings that the N-terminal fragment is sufficient for driving CAB localization. However, since the C-terminal coiled-coil domain of MCAK drives dimerization[43], it is not clear why the C-

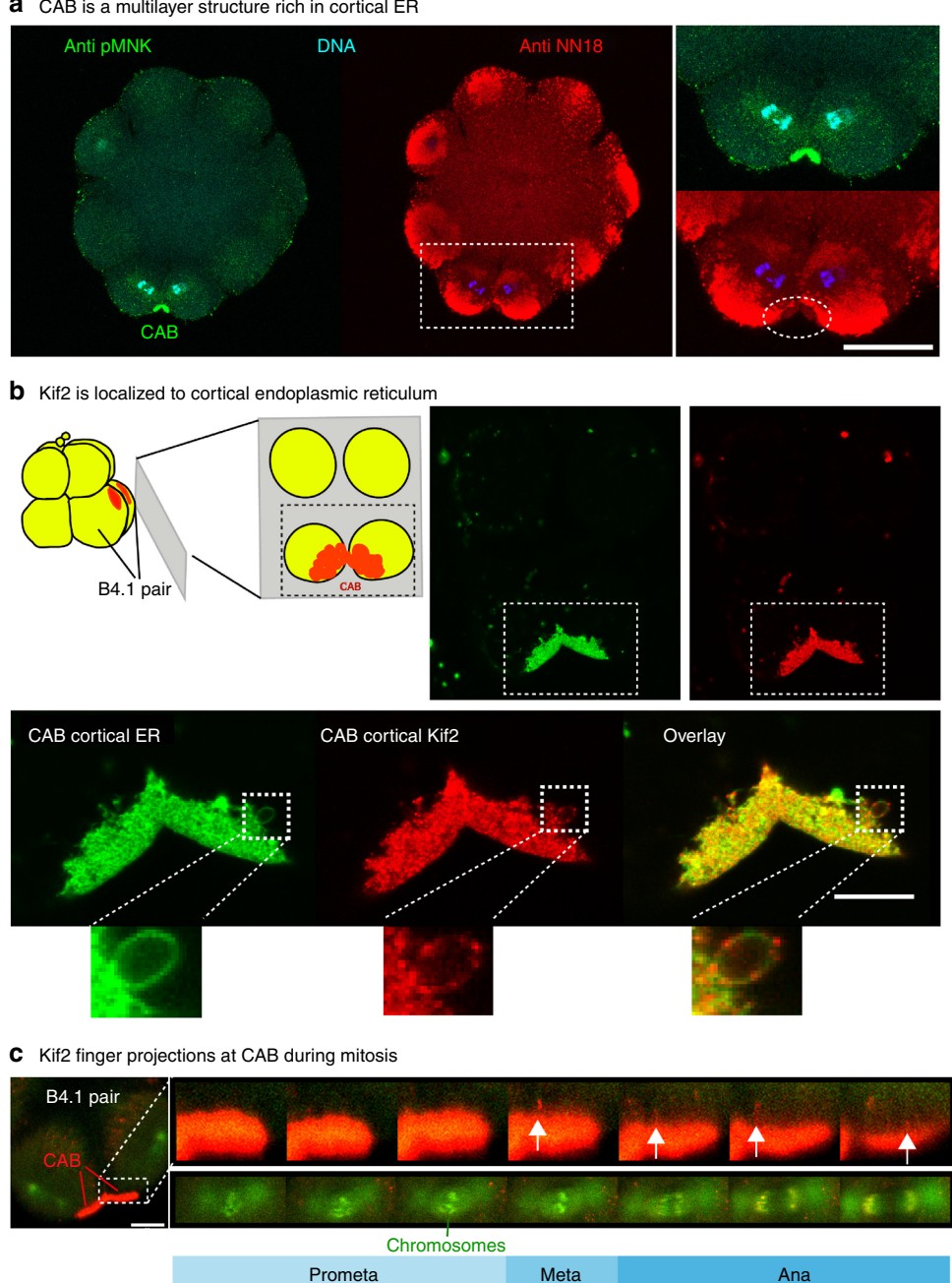

**a** CAB is a multilayer structure rich in cortical ER

Anti pMNK    DNA    Anti NN18

CAB

**b** Kif2 is localized to cortical endoplasmic reticulum

B4.1 pair    CAB

CAB cortical ER    CAB cortical Kif2    Overlay

**c** Kif2 finger projections at CAB during mitosis

B4.1 pair    CAB

Chromosomes

| Prometa | Meta | Ana |

**Fig. 4** Kif2 protein localizes to the domain of cortical ER in the CAB. **a** CAB is a multilayered cortical structure rich in cortical ER. pMNK labeling of the cortical ER (*green*) with the mitochondria labeled with anti NN18 (*red*) and the DNA with DAPI (*blue*). Enlarged view of the ROI showing that the mitochondria are excluded from the CAB where the cER is enriched (*oval*). *Scale bar* = 30 µm. *n* > 50. **b** Kif2 is localized to cortical endoplasmic reticulum. Schematic of cortical preparation, ER in *red*. *Top right*: probing cortical preparations with DiO (*green*) to label the endoplasmic reticulum and Kif2 antibody (*red*) revealed that Kif2 protein was localized to the domain of cortical ER in the CAB. CAB is indicated in boxed region. *Lower*: enlarged views of the boxed region in *top right* images showing more clearly the cER labeled with a DiO (*green*) together with the Kif2 labeling (*red*). At the edge of the CAB some tubes of cER are visible (*insets of small boxed regions*) where Kif2 fluorescence appears punctate relative to the green DiO cER signal. *Scale bars* = 10 µm. *n* > 50. See Supplementary Movie 9. **c** Kif2 finger-like projections at CAB surface during mitosis. Embryo containing Kif2::Tom, EB3::GFP and H2B::GFP. During mitosis red finger-like projections appear at the surface of the CAB (*arrows*). Mitotic stage is indicated from H2B::GFP labeling of chromosomes during mitosis (left boxed region). Enlarged views of the two boxed regions showing the finger-like projections from the CAB surface (*upper row* of images) and the mitotic stage as indicated by the chromosome configuration. *Scale bar* = 20 µm

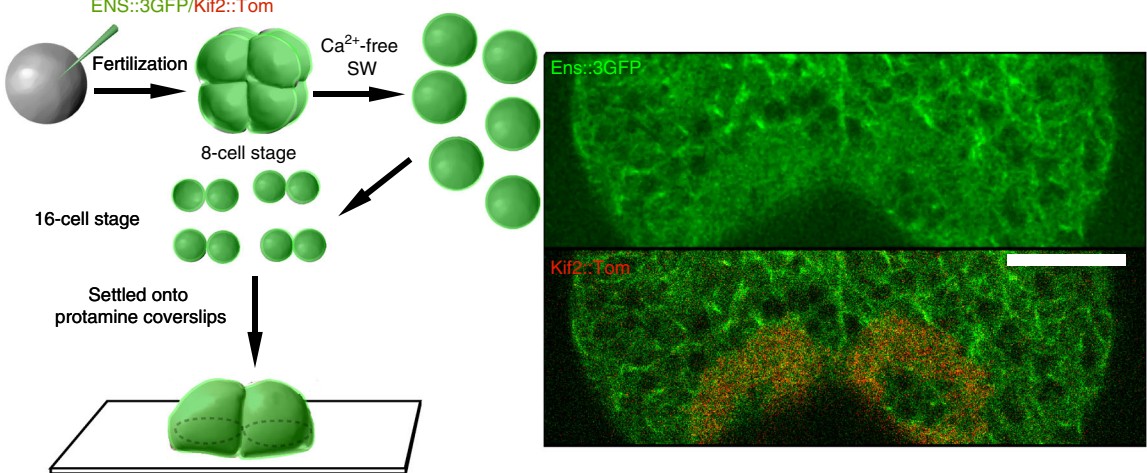

**a**  Microtubules are less abundant in the CAB

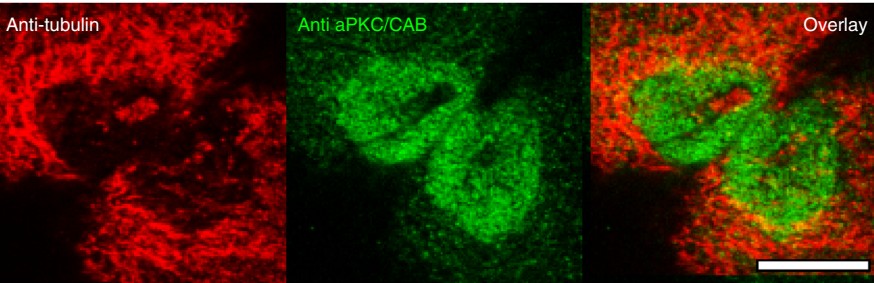

**b**  Microtubules are less abundant in the CAB

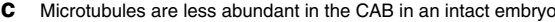

**c**  Microtubules are less abundant in the CAB in an intact embryo

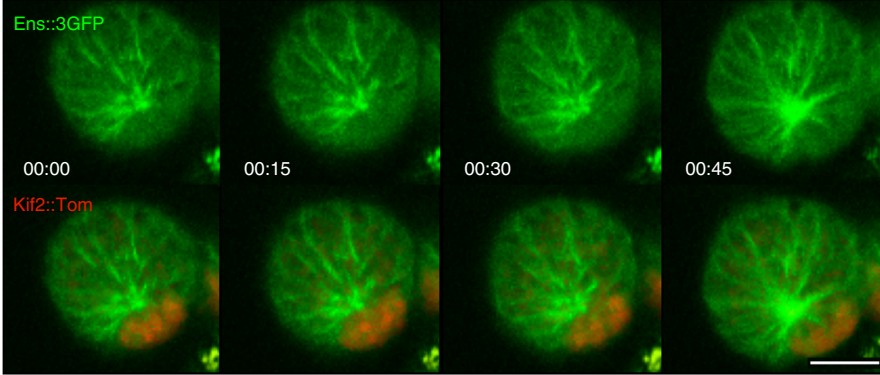

**Fig. 5** Microtubule dynamics at the cortical CAB. **a** Microtubules are less abundant in the CAB. In order to follow microtubule dynamics at the CAB, unfertilized eggs were microinjected with mRNA for Ens::3GFP and Kif2::Tom, fertilized and then transferred to calcium-free seawater at the 8-cell stage to dissociate the 8 blastomeres. Once the dissociated blastomeres had divided, pairs of B5.2 blastomeres were placed on coverslips that had been treated with protamine so that the blastomeres adhered. Next fast confocal imaging (0.8 s/image) of a 1 μm thick optical section just above the coverslip revealed that microtubules were less abundant in the CAB (red). *Scale bar* = 10 μm. *n* = 4. See Supplementary Movie 10. **b** Microtubules are less abundant in the CAB. Confocal z-sections from fixed 8-cell stage embryos showing microtubules (*red*, anti-tubulin) and the CAB (*green*, anti-aPKC). Microtubules are less abundant in the CAB (*n* > 50). *Scale bar* = 20 μm. **c** Microtubule dynamics at the CAB in an intact embryo. Selected frames from a confocal time-lapse series through the surface of a CAB-containing blastomere at the 32-cell stage. Microtubules are labeled with Ens::3GFP and the CAB with Kif2::Tom. Confocal images were acquired every 3 s and selected frames from each are shown. Microtubules are less abundant in the CAB (Supplementary Movie 11). *Scale bar* = 10 μm

terminal domain of ascidian Kif2 does not localize to the CAB by forming a dimer with endogenous Kif2 in the CAB.

**Kif2 is lost from the CAB after NEB.** We noticed that Kif2 protein appeared to accumulate at the CAB during interphase and leave the CAB during mitosis (Supplementary Movie 7). In order

to determine more precisely the dynamics of Kif2 delocalization from the CAB we performed live-cell ratiometric imaging of Kif2::Tom levels relative to Plk1::Ven, which is also a CAB-resident protein (Fig. 3a). By comparing the fluorescence of Kif2::Tom to Plk1::Ven we found that Kif2::Tom began to be lost from the CAB within 1 min. of NEB (*n* = 9, 42 s. ± 17, mean ± s.e.m.) and continued to fall throughout prometaphase (Fig. 3a and

Supplementary Movie 8). Careful examination of endogenous Kif2 levels at the CAB in either *Phallusia* or *Ciona* embryos revealed a similar dynamic delocalization of Kif2 during mitosis, whereby endogenous KIF2 protein was delocalized from the CAB by anaphase (Fig. 3b and Supplementary Fig. 3). Since we scored the immunofluorescence data shown in Fig. 3b for presence or absence of Kif2 in the CAB, it should be noted the bars at metaphase appear high because a small amount of Kif2 remains localized to the CAB at that time. However, these data reflect presence or absence of Kif2 in the CAB and not the absolute amount of Kif2 which is difficult to estimate quantitatively from immunofluorescence data. Moreover, we are confident that we

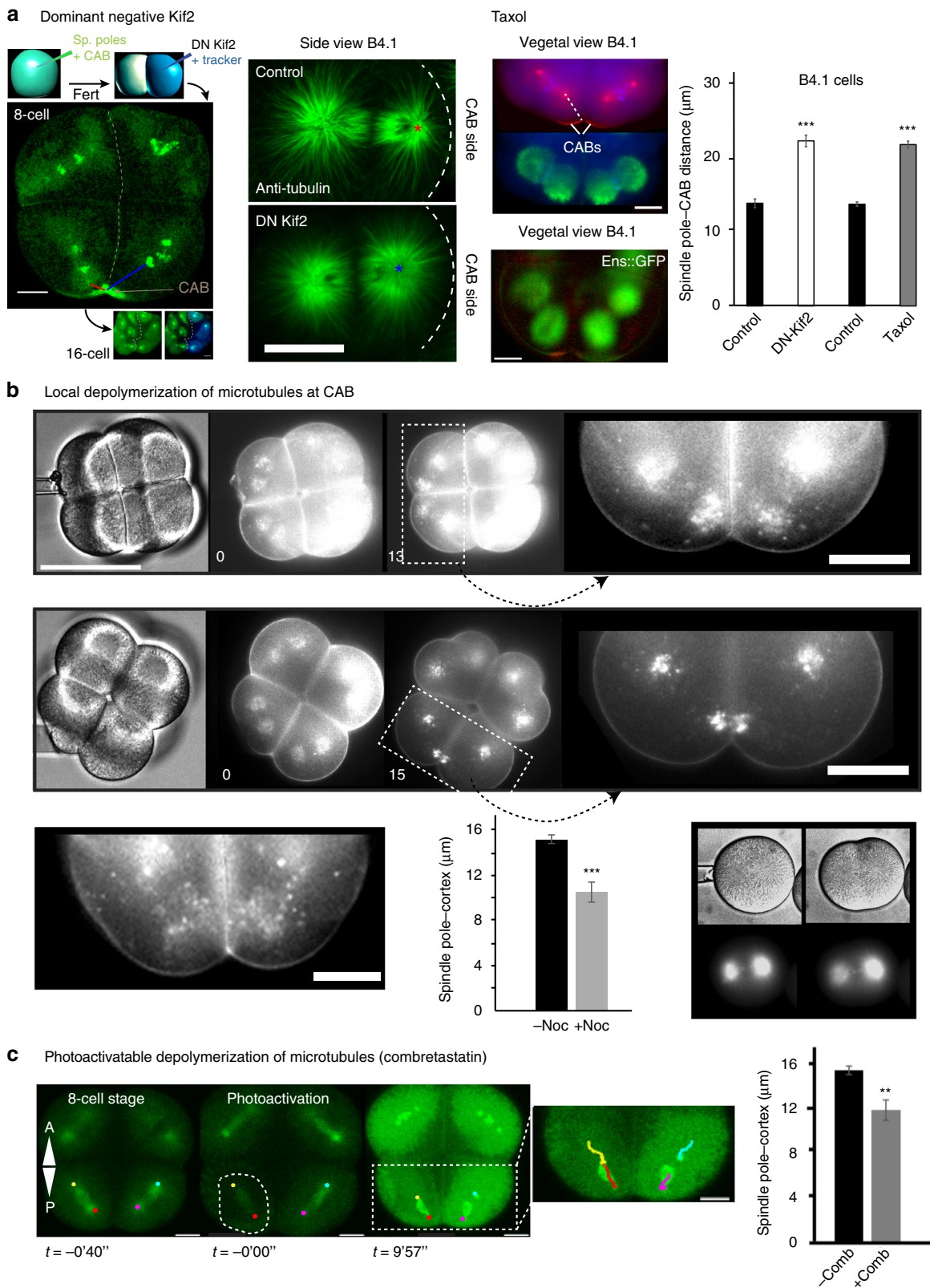

can detect the loss of endogenous Kif2 from the CAB since we found that two other CAB-resident proteins such as PEM1 or aPKC do not leave the CAB during mitosis (Fig. 3b).

In order to determine whether loss of signal at the CAB was due to delocalization or degradation we used the photoconvertible construct Kif2::Dendra to follow a specific pool of Kif2 protein in vivo. UV illumination converts the green Kif2::Dendra fusion protein into red Kif2::Dendra. UV illumination of Kif2::Dendra in a region within one CAB caused just the illuminated CAB to become red (Fig. 3c, *left panel arrowheads*). Since Kif2 is also a kinetochore localized protein, we reasoned that if we could detect red Kif2::Dendra on chromosomes this would indicate that the red version of Kif2 protein diffused from the nearby CAB to become captured by the adjacent chromosomes. It is important to note that the presence of red Kif2::Dendra on chromosomes does not rule-out destruction of Kif2 at the CAB, but it does show that some red Kif2 protein is capable of leaving the CAB intact. In addition, since ascidian Kif2 lacks a destruction box motif it is therefore unlikely to be a substrate of the anaphase-promoting complex/cyclosome which targets protein like cyclins A and B for destruction during M phase as we showed in the ascidian[44, 45]. Figure 3c *right panel* shows that we were able to detect red Kif2::Dendra on chromosomes in the blastomere containing the photo-converted red Kif2::Dendra. Note that on the *left panel* in Fig. 3c the blastomeres are in interphase and Kif2::Dendra does not label the decondensed DNA. Following photoconversion, some of the red version of Kif2::Dendra that was created in the CAB (Fig. 3c, *right panel arrowhead*) has diffused away from the CAB to label nearby chromosomes (Fig. 3c, *right panel, arrows*). Since the red version of Kif2 that was created in the CAB could be detected on the chromosomes we concluded that Kif2 protein can delocalize from the CAB.

**Kif2 localizes to cortical ER in the CAB**. As noted previously, the CAB is a multilayered structure comprised of a thick layer of cER protruding into the cytoplasm which adheres to a specialized region of actin-rich cortex[36]. By co-staining for mitochondria which surround and outline the cER mass, and for pMNK, a cER resident protein[37], the deeper cER can be visualized (Fig. 4a). Unlike aPKC protein, which labels the CAB cortex (Fig. 1b), Kif2 protein occupied the thicker cER layer (Fig. 4a). To determine the precise localization of Kif2 protein in the CAB we prepared isolated cortices. By sticking 8-cell stage embryos to coverslips followed by shearing using an isotonic buffer, the cortex and its associated cER is retained on the coverslip (Fig. 4b). Labeling these cortical preparations with DiO, an endoplasmic reticulum marker in isolated cortices[37, 46] (Supplementary Fig. 6), followed by anti-Kif2 revealed a concentration of Kif2 on the cER of the CAB (Fig. 4b, upper panel, and lower panel enlarged view for higher definition). In Fig. 4b a tube of cER extruding from the CAB labeled with DiO has a punctate staining pattern for Kif2 (*inset*: enlarged view in Fig. 4b lower panel and Supplementary Movie 9). In live embryos we were able to observe dynamic finger-like projections from the surface of the CAB labeled with Kif2::Tom (Fig. 4c, *arrow*), consistent with the notion that Kif2 was localized to the tubes of cER in the CAB.

**Microtubule dynamics at the CAB**. To visualize the rapid dynamics of microtubules at the cortex embryos have to be immobilized close to a flat surface, here provided by the coverslip. In one cell *C. elegans* zygotes this permitted the accurate determination of plus end dynamics leading to the development of the microtubule "touch and pull" mechanism of ACD[13]. In ascidians, measurement of microtubule plus end dynamics in a cortical slice containing the CAB and non-CAB cortex is complicated by both the movement and the geometry of the embryo: since the CAB is an apical cortical structure close to the midline it invariably curves away from the coverslip. In order to overcome these problems, we dissociated blastomeres with calcium-free seawater and used protamine-coated coverslips to immobilize the blastomeres (Fig. 5a). Blastomere isolation does not perturb UCD in *Phallusia* embryos[34]. Ens::3GFP fluorescence was then measured in a cortical slice within 1 μm of the coverslip every 0.8 s (Fig. 5).

**Fig. 6** Inhibition of Kif2 and localized microtubule depolymerization. **a** Dominant negative Kif2. *Left*: 8-cell stage embryo in which one blastomere at the 2-cell stage was microinjected with a truncated version of Kif2 which acts as a dominant negative (henceforth DN-Kif2) and histone H2B::Rfp1 as a fluorescent tracker to label the injected half of the embryos. Unfertilized eggs had previously been injected with Ens::3GFP to monitor spindle poles and Kif2 Nter::Ven to monitor the CAB. Minimum spindle pole distance to the CAB was measured at cleavage onset in the control half of the embryo (*blue line*) and the half containing DN-Kif2 (*red line*). *Middle*: Embryos injected at the 2-cell stage with DN-Kif2 were fixed and labeled for microtubules. A side view of an 8-cell stage embryo shows that the CAB-proximal aster is larger in the presence of DN-Kif2, and the spindle pole distance to the CAB cortex is increased (*stars* represents the center of the spindle pole closest to the CAB). Taxol: Treating embryos at the 8-cell stage with Taxol to stabilize microtubules also created asters that appeared more equal in diameter and increased the spindle pole-CAB distance. *Top*: epifluorescence images of fixed 8-cell stage embryo. *Upper image* showing spindle poles and the CAB (anti-gamma tubulin/anti-aPKC), *lower image* microtubules (anti-Tubulin). *Dotted line* shows measured distance. n = 30. *Lower*: epifluorescence image of live 8-cell stage embryo treated with Taxol showing merged images (microtubules *green*, CAB *red*). Scale bars = 20 μm. *Right*: quantification of spindle pole to CAB distance for DN-Kif2 and Taxol versus wild-type embryos. For the DN-Kif2 experiment the spindle pole to CAB distance was 13.6 ± 0.59 μm (mean ± s.e.m.) for wild type versus 21.6 ± 0.5 μm (mean ± s.e.m,) in the presence of DN-Kif2. Student's *t*-test, ***$P$ = < 0.00005. n = 13. For the Taxol experiment the spindle pole to CAB distance was 13.7 ± 0.3 μm (mean ± s.e.m.) for wild type versus 21.6 ± 0.5 μm (mean ± s.e.m) in the presence of Taxol. Student's *t*-test, ***$P$ = < 0.00005. n = 30. **b** Local depolymerization of microtubules at CAB. *Top row* of images. Nocodazole pipette (small bore) was advanced towards one B4.1 blastomere (bright-field image time 0). Spindle position during mitosis (13 min. image). *Zoom* of boxed region showing the spindle pole closest to the pipette moved even closer towards the CAB during mitosis (compare left spindle with right spindle pole). See Supplementary Movie 13. Scale bar, 30 μm. *Middle row* of images. Nocodazole pipette (large bore) was advanced towards one B4.1 blastomere (bright-field image time 0). Fluorescence image showing spindle position during mitosis (15 min. image). *Zoom* of boxed region showing that both spindle poles moved closer to the CAB and midline (*arrows*). Scale bar, 30 μm. *Lower left* panel and quantification. Control blastomeres at 8-cell stage. Spindle-pole–cortex distance in the presence of the nocodazole pipette was 10.5 ± 0.9 μm (mean ± s.e.m, n = 16) versus 15.2 ± 0.4 μm (mean ± s.e.m, n = 32) without the pipette. Student's *t*-test, ***$P$ = < 0.00005. *Right panel*. Nocodazole pipettes were tested on fertilized eggs containing Ens::3GFP. Note the loss of microtubule density in the aster nearest the pipette. Time in min. Scale bar, 30 μm. See Supplementary Movie 12. **c** Microtubule depolymerization with the caged Combretastatin. Caged Combretastatin was uncaged causing its activation in the boxed region causing microtubule depolymerization during mitosis. Microtubules were labeled with Ens::3GFP. Spindles were left intact and still migrated towards the CAB. Due to diffusion of the uncaged Combretastatin all cells were affected. Tracking of spindle poles is shown in the inset to the right. Scale bars = 20 μm Spindle-pole-cortex distance following uncaging of Combretastatin was 11.7 ± 0.9 μm (mean ± s.e.m, n = 16) versus 15.2 ± 0.4 μm (mean ± s.e.m, n = 32) without the Combretastatin. Student's *t*-test, **$P$ = < 0.005. n = 16

Microtubules are present as a dense network on the cortex during interphase, but in the CAB domain they are less abundant (Fig. 5a, b, see Supplementary Movie 10). In Supplementary Movie 10, microtubules can be observed polymerizing in the direction of the CAB (labeled red with Kif2::Tom), but they penetrate the CAB less frequently. Occasionally during interphase the CAB has a hole at its center where microtubules can be observed to reach the cortex, but microtubules are less abundant in the Kif2-labeled zone of the CAB (Fig. 5a, b). Microtubules are also less abundant in the CAB in the intact embryo at the 32-cell stage even though they can be seen growing around the CAB (Fig. 5c and Supplementary Movie 11) as in isolated blastomeres (Fig. 5a); however, since the embryo moves in the imaging plane many z sections were acquired so the time resolution in one z plane is reduced.

**Microtubule depolymerization and aster asymmetry.** In order to determine the role of Kif2 in the CAB we generated a mutant form of Kif2 protein (DN-Kif2) based on the construct which behaves as a dominant negative in mammalian cells[17]. DN-Kif2 contains the N-terminal domain which targets it to the CAB and a C-terminal domain which permits dimerization of MCAK in mammalian cells[43], but lacks the catalytic domain (Fig. 6). mRNA encoding DN-Kif2 and H2B::Rfp1 constructs were injected into one blastomere of a two cell stage embryo previously injected with Ens::3GFP and Kif2-Nter::Venus to monitor microtubules and the CAB in both halves of the embryo (Fig. 6a). As a control we compared the effect of DN-Kif2 to wild-type Kif2 by co-injecting eggs with the same concentration of mRNA encoding either DN-Kif2 or wild-type Kif2 together with the same concentration of Ens::3GFP to monitor fluorescence. In order to distinguish those eggs injected with a mixture of wild-type Kif2 plus Ens::3GFP from those injected with DN-Kif2 plus Ens::3GFP (since both batches will display green fluorescence), we mixed low levels of H2B::mRFP1 mRNA with DN-Kif2/Ens mRNA before microinjection (DN-Kif2 eggs thus also display red fluorescence). All eggs were fertilized shortly after microinjection and monitored up to the 32-cell stage. None of the DN-Kif2 injected eggs reached the 32-cell stage ($n = 32$) while 12/16 wild-type Kif2 reached the 32-cell stage (Supplementary Fig. 7). DN-Kif2 prevented the development of aster asymmetry (Fig. 6a, *right panel*). We measured the distance between the nearest spindle pole and the CAB in both halves of the embryo at the 8-cell stage and found that DN-Kif2 significantly increased the spindle pole to CAB distance (Fig. 6a). Since taxol stabilizes microtubules and reduces depolymerization, we reasoned that taxol should have a similar effect to DN-Kif2. Treating embryos in mitosis with taxol also prevented the development of aster asymmetry and increased the distance of the spindle pole to the cortex and CAB (Fig. 6a, *middle panel*). Either Kif2-DN or Taxol treatment increased the spindle pole to CAB distance significantly (Fig. 6a, *right panel*). These data suggest that aster asymmetry and asymmetric spindle positioning require the activity of Kif2 and microtubule depolymerization.

If aster reduction facilitates the migration of one spindle pole towards the cortex, we reasoned that increasing microtubule depolymerization near the CAB would enhance spindle pole movement towards the cortex. We employed a method to depolymerize microtubules locally by using a micro-pipette as a spatially confined source of nocodazole (Materials and methods section) and monitored the loss of Ens::3GFP microtubule labeling near the source of the pipette (Fig. 6b, *lower right panel* and Supplementary Movie 12). We applied different diameter pipettes containing nocodazole near the CAB immediately after NEB and measured the effect on the position of the proximal

spindle pole. Relatively small diameter nocodazole pipettes caused the CAB-proximal spindle pole to migrate closer toward the midline cortex (Fig. 6b and Supplementary Movie 13). Blastomeres far from the source of nocodazole were unaffected and divided normally (Fig. 6b, *top panel* and Supplementary Movie 13). By using a larger bore pipette we could affect both CAB-containing blastomeres so that both CAB-proximal spindle poles moved closer to the midline cortex while blastomeres farther from the pipette behaved normally and divided (Fig. 6b, *middle panel*). Control dimethyl sulfoxide (DMSO)-containing pipettes had absolutely no effect on spindle positioning (Fig. 6b, *bottom left*). Placing nocodazole pipettes on non-CAB blastomeres did not cause spindle poles to move towards the nocodazole needle and midline (Supplementary Fig. 8). To depolymerize microtubules via a second method we used photo-activation of caged Combretastatin[47] while imaging microtubules with Ens::3GFP or EB3::GFP (Fig. 6c). Activation of caged Combretastatin during mitosis caused astral microtubule depolymerization while leaving spindle microtubules intact. As noted with nocodazole pipettes, the whole spindle migrated even closer towards the midline cortex (Fig. 6c). These data revealed that reducing aster size could augment the asymmetric positioning of the spindle, supporting the hypothesis that astral microtubule polymerization opposes the pulling forces provided by the CAB.

**Discussion**

Here we present evidence for the localization and function of a microtubule depolymerase (Kif2) at a cortical site that is involved in asymmetric spindle positioning. Precise control of microtubule dynamics at the cortex is a fundamental cellular mechanism that is involved in diverse biological processes ranging from the control of axon morphology[6, 7, 48] to the unequal division of cells[4, 8]. UCD is an extremely widespread process in biology occurring in bacteria[49, 50], yeast[51], and many different types of embryo. Amongst embryos, UCD has been observed in ctenophores[52], chaetognaths[53], spiralians[30], echinoderms[31], and invertebrate chordate embryos of the ascidian[33]. A common theme among several embryos that display UCD is the development of aster asymmetry. Unequally cleaving spiralian embryos (*Tubifex*, and the leech *Helobdella*) display aster asymmetry at the 1–2-cell stage leading to eccentric positioning of the mitotic spindle[39]. Sea urchin embryos also display UCD starting at the 8–16-cell stage culminating in the formation of four micromeres[31] at the vegetal pole where aPKC is absent[54]. Although the mechanism underlying this UCD is not known in these embryos, observations indicate that one centrosome becomes disk-shaped and closely apposed to the cortex before UCD[31]. In ascidians three successive rounds of UCD occur starting at the 8–16-cell stage, resulting in the formation of two small blastomeres at the 64-cell stage that are germ cell precursors[33]. We showed here (Fig. 1) and previously that the CAB causes one pole of the mitotic spindle to approach the CAB during prometaphase through anaphase which is accompanied by the shrinking of the CAB-proximal aster[34].

One central unresolved question therefore is how specialized cortical sites affect astral microtubules leading to the development of aster asymmetry. Here we report that Kif2 is concentrated on a subdomain of cER found at the CAB during interphase and leaves the CAB within 1 min of NEB (Fig. 3a and Supplementary Movie 8). Interestingly, it was recently found that in human cells Kif2A can also associate with organelles[55]. Here for example, Kif2A associates with a sub-type of Arf GAPs (AGAP1) that is found on endosomes[55] and this association between Kif2A and AGAP1 is involved in cytoskeletal remodeling and cell movement. In the ascidian, we propose that the Kif2 localized to the

cER affects microtubule dynamics at the cortex, and also following release from the cER causes depolymerization of the nearest microtubules during mitosis thus leading to the development of aster asymmetry. By artificially increasing the amount of local depolymerization of astral microtubules in the proximity of the CAB we found that the spindle pole moved even closer to the midline cortex (Fig. 6b).

Our results have led us to the conclusion that polymerization of astral microtubules opposes the pulling forces that likely displace the mitotic spindle towards the CAB. However, it is currently unknown how the CAB pulls the spindle towards the cortex[56] although this will be a key area for future studies. We therefore propose that the diffusion of Kif2 from the CAB in mitosis causes the local depolymerization of those astral microtubules nearest the CAB thus facilitating the eccentric positioning of the mitotic spindle near the CAB cortex. Furthermore, since MCAK can produce a significant pulling force during microtubule deploymerization[57] it is also possible that some of the remaining Kif2 localized at the CAB creates a pulling force as the microtubules touching the CAB depolymerize. The mechanism we have discovered here in a chordate deuterostome embryo extends our understanding of how the cortex is involved in asymmetric spindle positioning, which so far has been heavily studied in two protostomes, *C. elegans* and *Drosophila*. In *Drosophila* larval neuroblasts, even though the apically localized Gα/Pins/NuMA complex aligns the mitotic spindle[8, 58, 59], UCD can be driven by a cortical subdomain of Myosin II via a mechanism which is independent of spindle position[60]. Although neuroblasts display an alternative mode of UCD, work in *C. elegans* zygotes has demonstrated the fundamental importance of the conserved Gαi/GPR-1/2/LIN-5 (Gαi/Pins/NuMA) complex, which accumulates at the posterior cortex and creates a pulling force on astral microtubules that is greater at the posterior cortex, thus causing unequal cleavage[11, 61]. Intriguingly, microtubule depolymerization is thought to be required for UCD in *C. elegans*[13, 28, 62] although the mechanism responsible is currently unknown. It appears that this is not Kif2-dependent in *C. elegans*, since inhibition of MCAK with RNAi does not prevent eccentric spindle positioning[11].

The control of microtubule plus end dynamics by Kif2 that we find here is remarkably similar to the axonal pruning function of Kif2A in mammalian post-mitotic neurons. For example, in developing neurites Kif2A is thought to bind to the plasma membrane associated protein phosphatidylinositol 4-phosphate 5-kinase[48]. Kif2A localizes to the tips of developing neurites *in vivo* and functions to suppress collateral branch extension since the knockout of Kif2A in mice causes increased microtubule stability at the cell edge in growth cones creating more growth of collaterals[6].

Our results presented here indicate that the flattened aster with short microtubules and movement of the spindle toward the CAB are both facilitated by the action of a cortically localized microtubule depolymerase. Given that Kif2 is involved in asymmetric spindle positioning in invertebrate chordate embryos and morphogenesis of developing mammalian neurites, it may be worthwhile investigating the role played by kinesin-13 family members such as Kif2 and other microtubule depolymerases during UCD as well as other biological processes where microtubule dynamics alter when they encounter the cortex.

## Methods

**Origin of the animals**. *Phallusia mammillata* were collected at Sète (Etang de Tau, Mediterranean coast, France) and *Ciona intestinalis* at Roscoff. Ascidian eggs were dechorionated with 0.1% Trypsin (Sigma-Aldrich, T9201) in sea water for 90 min. then transferred to fresh sea water and stored until required. Ascidian sperm were activated with pH9.5 sea water for 10–20 min. then used to fertilize dechorionated eggs.

**Antibodies, fixation and reagents**. Embryos were fixed in −20° methanol containing 5 μM ethylene glycol-bis(β-aminoethyl ether)-N,N,N′,N′-tetraacetic acid (EGTA) and immunolabelled[46]. We used anti aPKC (1/200, Santa Cruz 216) and anti PEM1 (1/100)[37] to label the CAB, anti-tubulin (1/200, YL1/2 and 1/500 DM1a, Sigma) for microtubules, anti Kif2 (1/200 following affinity purification of anti-human Kif2[63] on *Phallusia* Kif2 protein produced in bacteria), and anti-γ tubulin (1/200, Sigma GTU88) for centrosomes. DiI (injection of saturated oil droplet into eggs, Invitrogen) and Mitotracker (2 μg/ml, Invitrogen) were used to label ER and mitochondria respectively in live embryos, Paclitaxel (4 μM, Sigma) to stabilize microtubules and nocodazole (20–50 μM, Sigma) to depolymerize microtubules. Caged Combretastatin (provided by Martin Wuhr) was added to 4-cell stage embryos at a final concentration of 100 nM and de-caged during 8-cell stage mitosis to depolymerize microtubules.

For Western blot samples were prepared in Laemmli sample buffer and migrated on 10% polyacrylamide gels by sodium dodecyl sulfate polyacrylamide gel electrophoresis using standard procedures. Two lanes were loaded containing either 40 uninjected eggs or 40 eggs which had been injected with mRNA encoding full-length Kif2::GFP. After transfer to nitrocellulose, each lane was cut into two strips length-wise which were incubated overnight with either anti-Kif2 or anti-GFP at a dilution of 1:1000 in TBS + 5% dry milk. The 4 strip blots were then washed in TBS + 0.1% tween, incubated with anti-rabbit secondary at 1:10,000, and washed in TBS-tween. The signal was detected using West Pico chemiluminescent substrate (Fisher) at an exposure of 2 min.

**Preparation of isolated cortices**. Isolated cortices were prepared following the methods previously established in the laboratory[36, 46]. Briefly, embryos at the desired stage were transferred to calcium-free seawater then placed at high density on coverslips coated with protamine (1 mg/ml) to which they adhere within 1 min. After two washes with cortex isolation medium (cortex isolation medium (CIM): 0.8 M glucose, 0.1 M KC1, 2 mM MgCl2, 5 mM EGTA, 10 mM MOPS buffer, pH 7), a stream of CIM is sprayed gently with a Pasteur pipette, shearing off the embryos but leaving attached to the glass imprints of the adherent membrane and associated cortical structures. The coverslips are washed rapidly in CIM then placed in cold methanol for fixation, then rehydrated in phosphate-buffered saline and processed for immunofluorescence by standard procedures used for whole embryos. ER of isolated cortices was labeled with the addition of 0.2 μg/ml DiO C6 (3) (Invitrogen) for 1 min. following fixation and immunolabelling[37, 46].

**Microinjection and imaging**. For microinjection dechorionated oocytes were mounted in glass wedges and injected with mRNA (1-2 μg/μl pipette concentration/~1–2% injection volume) using a high pressure system (Narishige IM300)[64]. mRNA-injected oocytes were left for 2–5 h or overnight before fertilization and imaging of fluorescent fusion protein constructs. The lipophilic dye Cell Mask Orange (Molecular Probes) was prepared at a concentration of 10 mg/ml in DMSO and diluted in sea water at 20 μg/ml then mixed 1:1 with the embryos just prior to imaging. Epifluorescence imaging was performed with an Olympus IX70, Zeiss Axiovert 100, or Axiovert 200 equipped with cooled CCD cameras and controlled with MetaMorph software package. Confocal microscopy was performed using a Leica SP5 or SP8 fitted with 40×/1.3NA oil objective lens and 40×/1.1NA water objective lens, respectively. All live imaging experiments were performed at 18–19 °C. For fast imaging of cortical preparations a rectangular image section of the imaging array was selected to increase the temporal resolution to 0.8 images/sec. Image analysis was performed using Image J, ICY and MetaMorph software packages. Calcium-free sea water: 450 mM NaCl, 9 mM KCl, 33 mM Na2SO4, 2.15 mM NaHCO3, 10 mM Tris pH 8, and 2.5 mM EGTA.

**Micromanipulation**. All manual micromanipulation experiments were performed on an Olympus IX70 microscope using a ×20 objective lens and Metamorph acquisition software. Embryos at the 4-cell stage were incubated with Cell Mask Orange diluted in sea water (1/1000) for 90 s then washed with sea water. Cell Mask labeled embryos were mounted at the 8-cell stage for observation. To prepare the nocodazole pipettes, nocodazole was added to liquid 1% low melt agarose in sea water giving a final concentration of 20–50 μM nocodazole. Using a Narishige PN30 puller microinjection pipettes were pulled from GC100-T glass (filament-free) capillary tubes. The tips of the microinjection pipettes were broken and calibrated by microscopic observation. These micropipettes were dipped into the liquid nocodazole/agarose solution, placed at room temperature which caused the agarose containing nocodazole to solidify. These prepared microinjection needles were stored in humid chambers and used the same day. The nocodazole/agarose needles were advanced towards the 8-cell stage embryos and placed on the surface of one B4.1 blastomere near the CAB starting at nuclear envelope breakdown. Bright-field and fluorescence images were acquired every 10–20 s using Meta-Morph software package.

**Synthesis of RNAs**. We used the Gateway system (Invitrogen) to prepare N-terminal and C-terminal fusion constructs using pSPE3::Venus (a gift from P. Lemaire), pSPE3::Rfp1, pSPE3::Cherry, pSPE3::tomato for all constructs except PH::GFP which was cloned into pRN3. For construct details please refer to our previous methods publication[38]. All Kif2 constructs were prepared using *Phallusia*

*mammillata* Kif2 (unique gene ID: phmamm.g00002556) and *Ciona intestinalis* Kif2 (unique gene ID: Ciinte.g00008837). All synthetic mRNAs were transcribed and capped with mMessage mMachine kit (Ambion).

**Bioinformatics and statistical analysis**. We created a database of animal protein sequences derived from the complete genomes of various metazoan lineages: *Amphimedon queenslandica*, *Mnemiopsis leidyi*, *Trichoplax adhaerens*, *Nematostella vectensis*, *Acropora digitifera*, *Hydra vulgaris*, *Crassostrea gigas*, *Aplysia californica*, *Capitella teleta*, *Lingula anatina*, *Caenorhabditis elegans*, *Drosophila melanogaster*, *Strongylocentrotus purpuratus*, *Branchiostoma floridae*, *Ciona intestinalis*, *Phallusia mammillata* and *Homo sapiens*. All data was retrieved from the NCBI genomes portal: https://www.ncbi.nlm.nih.gov/genome/browse/. *Phallusia mammillata* unique gene ID phmamm.g00002556 and *Ciona intestinalis* unique gene ID Ciinte.g00008837 were used here and throughout for creating all molecular tools.

We searched this database with the PFAM Kinesin hidden Markov model [PMID: 26673716] using the global local strategy implemented in HMMER2 [http://hmmer.org/download.html] and the model specific 'gathering threshold' bit scores as a cutoff. Kinesin regions (649 sequences), as defined by HMMER2 alignments, were extracted from full-length sequences and aligned using the MAFFT software package with default parameters [PMID: 12136088].

This alignment was used to create a phylogeny of all Kinesin domains, using the Bayesian approach implemented in phylobayes, with an LG + G model of sequence evolution [PMID: 19535536]. Two chains were run for 3500 generations. 700 generations were discarded as burnin. Although the chains had not converged, the region of the phylogeny around the human Kif2 proteins revealed a stable clade composed of orthologs of the human KIF19, KIF18, KIF24, and KIF2A/B/C genes with full support from posterior probabilities. The sequences representing this clade of orthologous groups were extracted. The *Phallusia mammillata* Kif13 protein sequence was added to the alignment, and another phylogeny reconstructed using the Phyml package with an LG + G evolutionary model and 100 bootstrap replicates [PMID: 20525638]. This tree is shown in Supplementary Fig. 2.

Sample sizes vary between experiments. No statistical methods were used to predetermine sample size. Statistical significances were assessed using unpaired two-tailed Student's *t*-test. *F*-test was used to determine variance and Kurtosis to determine skewness.

**Data availability**. All reagents are relevant data are available upon request.

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

## Acknowledgements

We thank Duane Compton and Bernardo Orr for the generous gift of anti Kif2 antibody, 505 Linda Worderman for useful advice, Remi Dumollard for thoughtful insights, and members of the LBDV for helpful discussions. We thank Mafalda Loreti and Paul Stolz for preliminary experiments that do not appear in the article. This work was supported by the Agence National de la Recherche (ANR-12-BSV2-0005-01), and in part by support from Sorbonne Universites ANR-11-IDEX-0004-02 to the Picard Network, and by NSF grant 124425 to D.B.

## Author contributions

A.D., J.C., and V.C. conceived project and conducted experiments. A.D. and J.C. wrote the ms. V.C., G.P., D.B. and M.F. performed experiments. R.R.C. performed bioinformatics analysis. C.H. and L.B. provided molecular tools.

## Additional information

**Competing interests:** The authors declare no competing financial interests.

