## [Peer Review File · Nature Communications]

Reviewers' Comments:

Reviewer #1 (Remarks to the Author)

Review of manuscript NCOMMS-16-18751-T by Costache et al.

The authors investigated mechanisms mediating unequal cell division in early ascidian embryos. They established that the kinesin microtubule depolymerase Kif2 is enriched in the centrosome attracting body (CAB), a structure located at the cell cortex and towards which the mitotic spindle is positioned during unequal embryonic divisions. The authors found that microtubules are depleted from the CAB. Furthermore, they demonstrated that injection of a dominant negative Kif2 construct prevented spindle pole movements towards the CAB. Experimental perturbations of microtubule dynamics likewise affect these movements, compatible with a mechanism whereby the presence of Kif2 at the CAB locally depolymerizes microtubules, thereby provoking spindle asymmetry and unequal cell division.

This is an interesting study conducted in a model system that is ideally suited for analyzing unequal cell division. For the most part, the work is well executed and the data convincing. However, some aspects need to be clarified further before publication could be endorsed, as detailed below.

Main points:

1) The data reported in Figure 6, which represents the meat of the paper, deserves further attention. Whereas Figure 6A convincingly shows an impact of dominant negative Kif2 on the position of the CAB-proximal spindle pole, this is not the case of the data shown in Aii) or in Bi, Bii and Biii. The authors need to quantify these experiments in the same manner as those reported in Figure 6A. In addition, as an additional control, the authors should provide wild-type Kif2 protein to verify that the observed impact is not merely due to an elevation of Kif2 levels, perhaps through an effect on a partner protein. Also, it is unfortunate that the authors appear not to have analyzed the consequences of dominant negative Kif2 protein injection on unequal cell division per se, as opposed to merely on spindle pole position; the wording in the title and elsewhere should be adapted accordingly. Furthermore, please remove the asterisks that currently hide in part the data in panel Bi and Biii.

2) The model put forth by the authors appears to have a temporal glitch. Indeed, whereas Kif2 is enriched at the CAB until metaphase, diffusing away at anaphase, the spindle pole approaches the CAB starting already in prometaphase. Thus, the postulated effect mediated by Kif2 diffusing away from the CAB (as stated for instance on page 14) would seem to occur too late to cause the asymmetry observed as early as prometaphase. This point needs to be discussed and clarified further.

3) The authors state that Kif2 is enriched in the cER of the CAB (page 10 and Figure 4B). I fail to see the data supporting this contention. Instead, it is reported that Kif2A is enriched in the CAB, and so is the cER, but double labelling with Kif2A and cER would be needed to ascertain the posited colocalization.

More minor points:

4) Figure 5A and the related Movie S8 are somewhat puzzling, with microtubules being rather unusual. Also, whereas it is clear that microtubules are less numerous in the CAB-containing

region, writing that they are "absent", as stated in the manuscript, seems inaccurate. More prudent language should be utilized.

6) Figure 5C. Why are Ens3 and Kif2 shown with the same color? This is potentially confusing because one does not know with certainty where each individual fusion protein localizes.

7) Most movies are beautiful, but also sometimes difficult to follow with precision for someone not working with ascidians. The authors should consider labeling the most salient features in the movies, so that they can serve as efficient supporting information for all readers.

8) On page 8, the authors claim that Kif2 localizes to spindle poles and chromosomes, referring to Figure 3. Which panel of Figure 3 shows this? In Figure 3A, there appears to be some signal on spindle microtubules rather than on chromosomes or on spindle poles. Please clarify.

9) Given that the C-terminal stalk domain of kinesins is known to mediate dimerization in general (see Endow, 2010), it is surprising that the C-terminal fragment of Kif2 would not localize to the CAB through heterodimerization with endogenous Kif2. The authors should comment on this point.

10) Figure 1. The asymmetry in aster size in the B5.2 pair shown in this figure is not that apparent (in particular, the asters of the blastomere on the left seem to be out of focus). Also, the last line of the legend of Figure 1 states that the "... dark unlabelled zone... is filled with cER"; this is not shown in this particular panel and should thus not be stated as such there.

11) Figure 2. The presence of what appears to be a degradation product in the +/+ lane should be mentioned.

12) Figure 3. As controls for panel A (on the right), the authors used a total of 5 embryos expressing Par6::Ven or H2B::GFP; they should spell out how many of each were used, and also whether it is justified to merge the two data sets. The same comment holds for Perm1 and aPKC antibody stains in panel B (also on the right).

13) Figure 3C. The authors show two examples of what happens with photoactivatable Kif2, which seem to exhibit a different behavior, with the embryo on the right being the only one showing labelling of chromosomes. How many embryos were analyzed in total, and what fraction exhibited such chromosomal labelling? Also, the embryo on the right seems to have much stronger signal to start with -could it be that the fact that a signal is observed on chromosomes in this case simply reflects differences in overall expression levels?

14) The layout of Figure 4C is somewhat confusing; the way things stand now, one has the impression that the two rectangles represent the same location in the embryo (which I guess is not the case).

15) Whereas the Introduction was a pleasure to read, the rest of the manuscript would benefit from further editing/polishing.

16) Page 4: the authors should mention the nature of the protein limiting microtubule growth at the cortex of *C. elegans* embryos, and perhaps discuss how its mechanism of action may relate to that of Kif2.

17) Page 4: Colombo et al.; 2003 and Tsou et al.; 2003 also reported asymmetric distribution of GPR-1/2, and should be quoted in addition to Gotta et al.; 2003.

18) Page 5, last line: please spell out whether 20 microns correspond to the diameter of the CAB.

19) The references are not listed alphabetically (even though this is how they appear in the main text); please fix.

20) Pages 6/7: please clarify the sentence that begins on page 6 and continues on page 7, as the current wording is somewhat confusing.

21) Page 9, the parenthesis that begins with "(it is important...)" is misplaced and should be moved further down in this paragraph.

22) Page 12, line 7, typo: "than" instead of "that".

23) Page 16: the authors should indicate the dilutions that were used in the immunofluorescence experiments. Moreover, it was not clear to me in which experiment DiI and DiO had been utilized.

Reviewer #2 (Remarks to the Author)

This work addresses the molecular mechanisms controlling a series of unequal cell divisions in early ascidian embryos. The main conclusion of the work is that a microtubule depolymerase of the kinesin 13 family locally depolymerases astral microtubules, thereby facilitating the displacement of the spindle towards the smaller daughter cell. Overall the work is convincing (but see below), though I cannot assess whether the work will be of interest to a broad audience. To increase the potential impact of the work, the authors could test a possible functional link between the localisation of the microtubule depolymerase and the PAR complex, localized in the centrosome attracting body.

My main concern is in the nearly complete lack of even the most basic statistics or quantification in many experiments. Figure 6 is particularly lacking both. In general, the authors provide a single example of localisations in WT or manipulated contexts, without commenting on the frequency of this phenotype, or when quantified on its significance (eg Fig 6Ai). This should be corrected before publication in any journal.

In addition some evidence provided may need to be strengthened. For instance, to convince the readers of the cortical ER localization of the kif2 protein, could the authors show its colocalization with the cER marker pMNK? This would be a more direct evidence than the evidence currently provided. Also, in Figure 3C, what is the fraction of CAB Kif2 protein detected on the chromosomes during mitosis? It seems to be a very minor fraction, in which case degradation rather than relocation might be the major fate of kif2 during mitosis.

More minor points:

Figure 1B: why was the same colour used for aPKC and NN18?

Figure 2B could be shifted to supplemental figures.

Figure 2D: it would be nice to mention that the N-ter domain of Kif2 has previously been shown to influence its subcellular localization.

Finally, the form of the manuscript could be significantly improved. For instance, the layout of the figures could be made more professional (and more accurately referenced in the text). The text should be expurgated from very technical descriptions. To encourage potential readers, the abstract should in particular avoid excessive use of acronyms and symbols.

Reviewer #3 (Remarks to the Author)

In the manuscript by Costache and colleagues, they have used the ascidian, a classic model

system for asymmetric segregation of cellular contents during development, to explore the mechanism by which cortically induced microtubule polymerization is used to promote asymmetric spindle positioning. The authors have discovered, using an antibody against the mammalian kinesin-13, Kif2A, and also via live imaging of a cloned ascidian Kif2 construct fused to fluorescent proteins, that Kif2 associates with mitotic structures expected based on the mammalian kinesin-13 literature. Significantly, Kif2 also strongly associated in a cell cycle dependent manner with the ER-rich centrosome-attracting body (CAB) which is implicated in positioning of the spindle for asymmetric cell divisions during early cleavage in ascidians. The authors hypothesize that the appearance of a member of the kinesin-13 microtubule depolymerizing family on the CAB might reflect the mechanism by which the spindle asters become asymmetric during unequal cleavage as one centrosome approaches the CAB. They have performed two experiments to bolster this claim. They have applied nocodazole to a restricted region of the cortex near the CAB to experimentally induce the movement of the spindle toward the CAB. They have also engineered a dominant-negative Kif2 construct which lacks putative MT depolymerizing activity but still associates with the CAB and have recorded that the distance between the CAB and the CAB-proximal spindle pole increased suggesting that the asymmetric spindle localizing machinery was impaired, an observation that was phenocopied by taxol administration.

I am enthusiastic about this study for a number of reasons. First, the localization of kinesin-13 members in mammalian cells have been reported on membrane-bound organelles (namely lysosomes) however their function there is not well understood because kinesin-13s lack transport activity. The present manuscript describes a novel and compelling activity for membrane-bound kinesin-13 activity in controlling the position of microtubule structures by spatially adjusting microtubule length. The association of Kif2 with the ER-rich CAB is particularly intriguing because enzymatic activation of Kif2A has recently been reported by an Arf-GAP associated with endosomes (Luo et al. 2016 JBC). Thus, association with the CAB has the potential to also be activating for Kif2's enzymatic activity - similar to the classic idea of "cargo activation" for conventional kinesin. Second, the authors are correct in stating that asymmetric spindle positioning in other, more well-studied systems have consistently described a requirement for kinesin-13 family members and microtubule depolymerization without identifying a good molecular mechanism for spatially selective destabilization of microtubules within the spindle. It is known in systems such as *C. elegans*, that a kinesin-13 is required for asymmetric spindle positioning but whether it is generally needed or specifically required in a spatially significant way is wholly unknown. The regulated association of a kinesin-13 family member on the surface of the CAB provides us with visually arresting insight into how this occurs in a cytoplasmically specialized organism. This can provide impetus for examining this mechanism in other cells where kinesin-13 family members may be more subtly employed in selective MT depolymerization.

I would like to see a couple of questions addressed prior to publication:

1-The authors applied nocodazole to the cortex opposite the proximal spindle pole and convincingly measured a decrease in distance between the pole and the CAB. Since the CAB is an area of localized MT depolymerizing activity (at least at certain times), this adds potential redundancy to the experiment. Did the authors try this experiment on non-CAB containing blastomeres as well? In other words, is an attractive force from the CAB necessary in addition to regulation of MT length? Cortical dynein contributes localized pulling forces and has been implicated in both asymmetric and symmetric spindle positioning. Thus, one would think that dynein is present and capable of pulling spindles off center in both CAB-containing blastomeres and those which divide symmetrically. Yet, the results in Figure 6Bii suggest that other blastomeres are immune to nocodazole-dependent relocalization. To understand this issue in more detail I would also like some clarification on the timing of nocodazole application versus the timing with which Kif2 dissociates from the CAB. I believe that these questions can be addressed by clarification in the text.

2-Similarly, is it possible to rescue the dominant-negative effect of the expressed mutant Kif2

protein with locally applied nocodazole? If not, does this indicate that DN Kif2 eliminates both MT-depolymerizing activity and attractive pulling forces on the spindle toward the CAB? I am not expecting this experiment to be performed as a condition of publication, however, I would like to understand in more detail why this experiment is not present.

In conclusion, this manuscript presents a number of technically challenging studies performed in a comparative system - a primitive chordate - that presents uniquely informative specialized features. Investigation of the activities associated with the CAB presents an opportunity for researchers, who possess the skills to utilize this system, to experimentally explore the spatial regulation of MT length - which is a key feature of many general cellular processes.

Reviewers' comments:

We thank all three reviewers for a thorough analysis of the strengths and weaknesses of our article, and in particular for the many inciteful comments that have greatly strengthened the article.

Reviewer #1 (Remarks to the Author):

We thank the reviewer for an thoughtful and thorough analysis of our article.

Review of manuscript NCOMMS-16-18751-T by Costache et al.

The authors investigated mechanisms mediating unequal cell division in early ascidian embryos. They established that the kinesin microtubule depolymerase Kif2 is enriched in the centrosome attracting body (CAB), a structure located at the cell cortex and towards which the mitotic spindle is positioned during unequal embryonic divisions. The authors found that microtubules are depleted from the CAB. Furthermore, they demonstrated that injection of a dominant negative Kif2 construct prevented spindle pole movements towards the CAB. Experimental perturbations of microtubule dynamics likewise affect these movements, compatible with a mechanism whereby the presence of Kif2 at the CAB locally depolymerizes microtubules, thereby provoking spindle asymmetry and unequal cell division.

This is an interesting study conducted in a model system that is ideally suited for analyzing unequal cell division. For the most part, the work is well executed and the data convincing. However, some aspects need to be clarified further before publication could be endorsed, as detailed below.

Main points:

1) The data reported in Figure 6, which represents the meat of the paper, deserves further attention. Whereas Figure 6A convincingly shows an impact of dominant negative Kif2 on the position of the CAB-proximal spindle pole, this is not the case of the data shown in Aii) or in Bi, Bii and Biii. The authors need to quantify these experiments in the same manner as those reported in Figure 6A. In addition, as an additional control, the authors should provide wild-type Kif2 protein to verify that the observed impact is not merely due to an elevation of Kif2 levels, perhaps through an effect on a partner protein. Also, it is unfortunate that the authors appear not to have analyzed the consequences of dominant negative Kif2 protein injection on unequal cell division per se, as opposed to merely on spindle pole position; the wording in the title and elsewhere should be adapted accordingly. Furthermore, please remove the asterisks that currently hide in part the data in panel Bi and Biii.

We have presented a quantification of all data presented in Figure 6 (shown in Fig. 6 iii and legend page 32) :

“iii) Quantification of spindle pole to CAB distance for DN-Kif2 and Taxol versus wild-type embryos. For the DN-Kif2 experiment the spindle pole to CAB distance was $13.6 \pm 0.59 \mu\text{m}$ (mean \pm s.e.m.) for wild type versus $21.6 \pm 0.5 \mu\text{m}$ (mean \pm s.e.m.) in the presence of DN-Kif2. Student’s *t*-test, $***P < 0.00005$. $n=13$. For the Taxol experiment the spindle pole to CAB distance was $13.7 \pm 0.3 \mu\text{m}$ (mean \pm s.e.m.) for wild type versus $21.6 \pm 0.5 \mu\text{m}$ (mean \pm s.e.m.) in the presence of Taxol. Student’s *t*-test, $***P < 0.00005$. $n=30$.”

Also see new sentence on page 12 :

“Either Kif2-DN or Taxol treatment increased the spindle pole to CAB distance significantly (Figure 6 Aiii). “

Wild type Kif2 experiment has been detailed and the results now appear in sFig6 and in the text at page 11 :

“As a control we compared the effect of DN-Kif2 to wild type Kif2 by co-injecting eggs with the same concentration of mRNA encoding either DN Kif2 or wild type Kif2 together with the same concentration of ENS::3GFP to monitor fluorescence. We used low levels of H2B::mRFP1 mixed with Kif2/Ens to distinguish the two groups of injected eggs. All eggs were fertilized and monitored up to the 32-cell stage. None of the DN Kif2 injected eggs reached the 32-cell stage ($n=32$) while 12/16 wild type Kif2 reached the 32-cell stage (Supplementary Fig. 7). “

Unequal cell division has been changed to asymmetric spindle position throughout the manuscript.

2) The model put forth by the authors appears to have a temporal glitch. Indeed, whereas Kif2 is enriched at the CAB until metaphase, diffusing away at anaphase, the spindle pole approaches the CAB starting already in prometaphase. Thus, the postulated effect mediated by Kif2 diffusing away from the CAB (as stated for instance on page 14) would seem to occur too late to cause the asymmetry observed as early as prometaphase. This point needs to be discussed and clarified further.

We have altered the text to remove all confusion about the time course of Kif2 delocalization from the CAB. The source of confusion likely comes from our analysis of the fixed data. Our fixed data provided us with qualitative data with which to score for the presence or absence of Kif2 in the CAB, but we were not confident in using these data to measure the quantity of Kif2 in the CAB with the required temporal resolution (since batches of embryos are not synchronised enough). To determine when after NEB Kif2 levels started to decline in the CAB we therefore performed ratiometric imaging. Due to the reviewers comments we have since replaced the data in Figure 3A with ratiometric data showing the level of Kif2::Tom and Plk1 ::Ven in the CAB. Both Kif2 and Plk1 localize to the CAB (see Figure 3A part 1). The ratiometric analysis in Figure 3A shows that Kif2 ::Tom fluorescence starts to fall 1 min. after NEB while the Plk1 fluorescence remains more stable. We have also added supplementary movie 8 to illustrate these data. Figure legend 3 has been altered to reflect the changes.

The following sentences have been changed and added to page 8 in order to clarify this point :

“In order to determine more precisely the dynamics of Kif2 delocalization from the CAB we performed live cell ratiometric imaging of Kif2::Tom levels relative to Plk1::Ven which is also a CAB-resident protein. By comparing the fluorescence of Kif2::Tom to Plk1::Ven we found that Kif2::Tom began to be lost from the CAB within 1 min. of NEB (n=9, 42 sec. +/- 17, mean +/- s.e.m.) and continued to fall throughout prometaphase (Figure 3A and Supplementary Movie S8). “We scored the immunofluorescence data shown in Figure 3B for presence or absence of Kif2 in the CAB hence the bars at metaphase appear high since some Kif2 remains localized to the CAB. However, this reflects presence and not the absolute amount of Kif2.”

We have added a legend for this new supplementary Movie :

Movie S8. Kif2::Tom is lost from the CAB after NEB

“Supplementary Movie 8. Kif2 protein is lost from the CAB after NEB

Selected confocal images from a 4D time series of an 8-cell stage embryo containing Kif2::Tom (red) and Plk1::Ven (green). Two stacks from the 4D time lapse series are shown (Z plane 1 and Z plane 2). Plk1::Ven labels the CAB (Z plane 1, see arrows) and is also a convenient marker of cell cycle phases (see Z plane 2 images) since Plk1::Ven also accumulates in the nucleus during interphase, labels the chromosomes until metaphase then the spindle midzone at anaphase (last image). Following NEB (at 8:50) the Kif2::Tom fluorescence in the CAB falls relative to Plk1::Ven. The Kif2 fluorescence in the CAB has already diminished at 10:36 min. and continued to fall (see 11:29 min. image) relative to the Plk1::Ven fluorescence which stays relatively constant during this time.”

3) The authors state that Kif2 is enriched in the cER of the CAB (page 10 and Figure 4B). I fail to see the data supporting this contention. Instead, it is reported that Kif2A is enriched in the CAB, and so is the cER, but double labelling with Kif2A and cER would be needed to ascertain the posited colocalization.

A double labelling experiment has been performed to confirm the statement that Kif2 is enriched in the cortical ER. See new Figure 4. The text has been amended to reflect these new data. We used DiO to label the endoplasmic reticulum as previously (see Paix et al., Dev Biol. 2011 357: 211-26 and Sardet et al., Methods Mol Biol. 2011. 770:365-400). We have also included a supplementary figure to demonstrate the reticular nature of the cortical ER in an unfertilized egg since the cER becomes more compacted in the CAB and its reticular appearance is more difficult to discern. However, often we can see loops of cER that protrude from the CAB. We have therefore chosen to show one z-stack where one such ER loop has been highlighted to show the punctate labelling with Kif2 versus the uniform labelling with DiO. We have added the following sentences to the article at page 10 :

“Labelling these cortical preparations with DiO, an endoplasmic reticulum marker in isolated cortices^{37,46} (and Supplementary Fig. 6), followed by anti-Kif2 revealed a concentration of Kif2 on the cER of the CAB (Figure 4 B upper panel, and lower panel enlarged view for higher definition). In Figure 4 B a tube of cER extruding from the CAB labelled with DiO has a punctate staining pattern for Kif2 (arrow in Figure 4 B lower panel and Supplementary Movie 9).”

Also see new Supp. Movie 9 legend :

“Supplementary Movie 9. Kif2 localizes to cortical endoplasmic reticulum.

Isolated cortices were prepared, fixed and labelled for cortical endoplasmic reticulum (DiO, green) and with anti-Kif2 (red) at the 8-cell stage. Z stacks of confocal optical sections shows the labelling pattern of the cER (green) and Kif2 (red). Note that at the edge of the CAB some tubes of cER can be seen (one is highlighted by the arrow) which are stained homogenously with DiO while the Kif2 labelling is more punctate. Scale bar = 10µm.”

More minor points:

4) Figure 5A and the related Movie S8 are somewhat puzzling, with microtubules being rather unusual. Also, whereas it is clear that microtubules are less numerous in the CAB-containing region, writing that they are "absent", as stated in the manuscript, seems inaccurate. More prudent language should be utilized.

Figure 5A and Movie S8. We thank the author for pointing out the inappropriate use of language. « Absent » has been replaced with « **less abundant** » throughout the manuscript.

6) Figure 5C. Why are Ens3 and Kif2 shown with the same color? This is potentially confusing because one does not know with certainty where each individual fusion protein localizes.

Figure 5C. We have redone this experiment (which we initially performed when we did not have both colors of the constructs available) with red and green versions to show the data more clearly. See new Figure 5C and Supplementary movie 11.

7) Most movies are beautiful, but also sometimes difficult to follow with precision for someone not working with ascidians. The authors should consider labeling the most salient features in the movies, so that they can serve as efficient supporting information for all readers.

Movies have been labelled throughout for clarity.

8) On page 8, the authors claim that Kif2 localizes to spindle poles and chromosomes, referring to Figure 3. Which panel of Figure 3 shows this? In Figure 3A, there appears to be some signal on spindle microtubules rather than on chromosomes or on spindle poles. Please clarify.

The text detailing Kif2 localization has been corrected. Page 7 new sentence :

“Ascidian Kif2 also localizes to **centrosomes and spindle microtubules** (like Kif2a/Kif2b) (**Supplementary Fig. 3**) although CAB localization was strongest.”

9) Given that the C-terminal stalk domain of kinesins is known to mediate dimerization in general (see Endow, 2010), it is surprising that the C-terminal fragment of Kif2 would not localize to the CAB through heterodimerization with endogenous Kif2. The authors should comment on this point.

Heterodimerization of Kif2. The reviewer is correct that the C-terminal stalk of MCAK is required for dimerization. We have added a new citation to the reference list and have been more precise with our language. Please see the following sentence on page 8 :

“The N-terminal domain of MCAK is involved in subcellular targeting⁴³ which is consistent with our findings that the N-terminal fragment is sufficient for driving CAB localization. However, it is not clear why the C-terminal dimerization domain⁴⁴ alone is not sufficient for targeting in the ascidian.”

10) Figure 1. The asymmetry in aster size in the B5.2 pair shown in this figure is not that apparent (in particular, the asters of the blastomere on the left seem to be out of focus). Also, the last line of the legend of Figure 1 states that the "... dark unlabelled zone... is filled with cER"; this is not shown in this particular panel and should thus not be stated as such there.

We agree with the reviewer – this was due to all asters not being in the plane of focus. This is particularly difficult at this stage since the spindles are tilted with respect to the viewer. However, we have included a new figure that shows the difference in aster size more clearly. We have replaced the previous example with a new image in Figure 1.

11) Figure 2. The presence of what appears to be a degradation product in the +/+ lane should be mentioned.

Degradation product in Figure 2 has been mentioned. Since this figure has been moved to the supplementary section (supplementary Figure 5) the appropriate figure legend has been amended.

12) Figure 3. As controls for panel A (on the right), the authors used a total of 5 embryos expressing Par6::Ven or H2B::GFP; they should spell out how many of each were used, and also whether it is justified to merge the two data sets. The same comment holds for Perm1 and aPKC antibody stains in panel B (also on the right).

Figure 3 has been changed since we agree it was confusing. Instead of showing a mixture of data from Par6 and H2B we have pooled all the Plk1 data to perform the ratiometric analysis in Figure 3A which now replaces the previous data. Importantly, the conclusion remains the same that Kif2 leaves the CAB in prometaphase. From these analyses we have measured that Kif2 leaves the CAB within 1 min. of NEB (n=9). We analyzed the 16–32 cell stage in most detail and show these data in Figure 3A. The following text has been added to the manuscript, page 8 :

“We noticed that Kif2 protein appeared to accumulate at the CAB during interphase and leave the CAB during mitosis (Figure 3 A and Supplementary Movie 7). In order to **determine more precisely the dynamics of Kif2 delocalization from** the CAB we performed live cell ratiometric imaging of Kif2::Tom levels relative to Plk1::Ven **which is also a CAB-resident protein. By comparing the fluorescence of Kif2::Tom to Plk1::Ven we found that Kif2::Tom began to be lost from the CAB within 1 min. of NEB (n=9, 42 sec. +/- 17, mean +/- s.e.m.) and continued to fall throughout prometaphase (Figure 3 A and Supplementary Supplementary Movie 8).** “

13) Figure 3C. The authors show two examples of what happens with photoactivatable Kif2, which seem to exhibit a different behavior, with the embryo on the right being the only one showing labelling of chromosomes. How many embryos were analyzed in total, and what fraction exhibited such chromosomal labelling? Also, the embryo on the right seems to have much stronger signal to start with -could it be that the fact that a signal is observed on chromosomes in this case simply reflects differences in overall expression levels?

The reviewer is correct to point out the difference. In Figure 3C the image to the left shows an interphase embryo with no condensed chromosomes so we would not expect to observe Kif2 labelling of kinetochores since they are not yet formed. We have added a sentence to the text to clarify this point on page 9 :

“Note that no Kif2::Dendra is present on chromosomes on the image to the left in Figure 3C since these embryos are in interphase and the chromosomes are decondensed. Following entry into mitosis Kif2::Dendra is detectable on chromosomes in only the left blastomeres where Kif2::Dendra had been photoconverted.”

14) The layout of Figure 4C is somewhat confusing; the way things stand now, one has the impression that the two rectangles represent the same location in the embryo (which I guess is not the case).

We have reworked Figure 4C to illustrate that two different regions of the same embryo are depicted.

15) Whereas the Introduction was a pleasure to read, the rest of the manuscript would benefit from further editing/polishing.

We are pleased that the reviewer enjoyed the introduction and have now made many changes throughout the rest of the manuscript which we hope will improve its reading quality (all additional text is indicated in red).

16) Page 4: the authors should mention the nature of the protein limiting microtubule growth at the cortex of *C. elegans* embryos, and perhaps discuss how its mechanism of action may relate to that of Kif2.

Page 4. Protein limiting MT growth at the cortex in *C. elegans* has been discussed. We have changed the following text on page 3/4 :

“However, in cells that divide unequally it is still not known what causes astral microtubule plus end depolymerization at the cortex. In *C. elegans* one protein has been described (EFA-6) which limits cortical microtubule growth, however the knockdown of EFA-6 does not prevent UCD¹⁸.”

17) Page 4: Colombo et al.; 2003 and Tsou et al.; 2003 also reported asymmetric distribution of GPR-1/2, and should be quoted in addition to Gotta et al.; 2003.

Colombo et al and Tsou et al. references have been included. See page 4 :

... while NuMAs binding partner GPR-1/2 (Pins/LGN) becomes enriched at the posterior cortex during mitosis²¹⁻²³. “

18) Page 5, last line: please spell out whether 20 microns correspond to the diameter of the CAB.

CAB diameter has been added. See page 5, text has been added :

« Due the large dimensions of the CAB (circa 20µm at the 8-cell stage, see Paix et al 2011 and this article)”.

19) The references are not listed alphabetically (even though this is how they appear in the main text); please fix.

Reference list has been corrected.

20) Pages 6/7: please clarify the sentence that begins on page 6 and continues on page 7, as the current wording is somewhat confusing.

Pages 6/7, for clarity the sentence has been changed to (now page 6):

“During these three rounds of UCD one pole of the mitotic spindle is attracted to a cortical structure termed the centrosome-attracting body or CAB (Figure 1A). “

21) Page 9, the parenthesis that begins with "(it is important...)" is misplaced and should be moved further down in this paragraph.

Parenthesis has been moved. Due to other comments this paragraph has been re-written, see page 9 :

“It is important to note that the presence of red Kif2::Dendra on chromosomes does not rule-out destruction of Kif2 at the CAB, but it does show that some red Kif2 protein is capable of leaving the CAB. However, since ascidian Kif2 lacks a destruction box motif it is therefore unlikely to be a substrate of the anaphase-promoting complex/cyclosome which targets protein like cyclins A and B for destruction during M phase as we showed in the ascidian^{45,46}”. Figure 3 C right panel shows that we were able to detect red Kif2::Dendra on chromosomes in the blastomere containing the photo-converted red Kif2::Dendra. Note that no Kif2::Dendra is present on chromosomes on the left panel in Figure 3 C since these embryos are in interphase and the chromosomes are decondensed. Following entry into mitosis (right panel in Figure 3 C) Kif2::Dendra is detectable on chromosomes in the blastomeres where Kif2::Dendra had been photoconverted. Thus the red Kif2 protein we detected on the chromosomes came from the CAB (Figure 3 C), indicating that some Kif2 protein leaves the CAB in mitosis.”

22) Page 12, line 7, typo: "than" instead of "that".

Typo than changed to that.

23) Page 16: the authors should indicate the dilutions that were used in the immunofluorescence experiments. Moreover, it was not clear to me in which experiment DiI and DiO had been utilized.

Dilutions for immunofluorescence exp. Have been added. See Materials section.

Clarification of DiI and DiO have been added. See page 17 sentence :

“As previously described^{37,59}, following fixation and before immunolabelling ER of isolated cortices was labelled with the addition of 0.2µg/ml DiO C6(3) (Invitrogen) for 1 min.”

And page 16

“DiI (injection of saturated oil droplet into eggs, Invitrogen) and Mitotracker (2µg/ml, Invitrogen) were used to label ER and mitochondria respectively in live embryos”.

Reviewer #2 (Remarks to the Author):

This work addresses the molecular mechanisms controlling a series of unequal cell divisions in early ascidian embryos. The main conclusion of the work is that a microtubule depolymerase of the kinesin 13 family locally depolymerases astral microtubules, thereby facilitating the displacement of the spindle towards the smaller daughter cell. Overall the work is convincing (but see below), though I cannot assess whether the work will be of interest to a broad audience. To increase the potential impact of the work, the authors could test a possible functional link between the localisation of the microtubule depolymerase and the PAR complex, localized in the centrosome attracting body.

We thank the reviewer for rigorously reviewing our manuscript.

We have performed the suggested experiment by exploring the functional link between Kif2 localization/delocalization to the CAB with the presence of the PAR complex at the CAB. Inhibition of aPKC activity with a pseudosubstrate did not affect either Kif2 accumulation or release from the CAB. Since our article was focussed on the role played by Kif2 we do not propose to add these new data to the manuscript.

My main concern is in the nearly complete lack of even the most basic statistics or quantification in many experiments. Figure 6 is particularly lacking both. In general, the authors provide a single example of localisations in WT or manipulated contexts, without commenting on the frequency of this phenotype, or when quantified on its significance (eg Fig 6Ai). This should be corrected before publication in any journal.

Statistical analysis of all data in Figure 6 has been provided. Please refer to Figure 6 and the legend on page 32 :

“iii) Quantification of spindle pole to CAB distance for DN-Kif2 and Taxol versus wild-type embryos. For the DN-Kif2 experiment the spindle pole to CAB distance was $13.6 \pm 0.59 \mu\text{m}$ (mean \pm s.e.m.) for wild type versus $21.6 \pm 0.5 \mu\text{m}$ (mean \pm s.e.m.) in the presence of DN-Kif2. Student's *t*-test, $***P < 0.00005$. $n=13$. For the Taxol experiment the spindle pole to CAB distance was $13.7 \pm 0.3 \mu\text{m}$ (mean \pm s.e.m.) for wild type versus $21.6 \pm 0.5 \mu\text{m}$ (mean \pm s.e.m) in the presence of Taxol. Student's *t*-test, $***P < 0.00005$. $n=30$.”

In addition some evidence provided may need to be strengthened. For instance, to convince the readers of the cortical ER localization of the kif2 protein, could the authors show its colocalization with the cER marker pMNC? This would be a more direct evidence than the evidence currently provided. Also, in Figure 3C, what is the fraction of CAB Kif2 protein detected on the chromosomes during mitosis? It seems to be a very minor fraction, in which case degradation rather than relocation might be the major fate of kif2 during mitosis.

We have performed a double labelling experiment to confirm the statement that Kif2 is enriched in the cortical ER. See new Figure 4, part B. The text has been amended to reflect these new data, see page 10:

“Labelling these cortical preparations with DiO, an endoplasmic reticulum marker in isolated cortices^{37,44} and Supplementary Fig. 6, followed by anti-Kif2 revealed a concentration of

Kif2 on the cER of the CAB (Figure 4 B, and inset for higher definition). In Figure 4 B a tube of cER extruding from the CAB labelled with DiO has a punctate staining pattern for Kif2 (arrow in Figure 4 B inset and Supplementary Movie 9).”

We have also included Supplementary Figure 6 to demonstrate the reticular nature of the cortical ER in an unfertilized egg. See New Supplementary Figure 6 and new Supplementary Figure 6 B legend, page 32. :

“(B) Kif2 is localized to cortical endoplasmic reticulum.

Schematic of cortical preparation, ER in red. Probing cortical preparations with Kif2 antibody and DiO to label the endoplasmic reticulum revealed that Kif2 protein (red) was localized to the domain of cortical ER (green) in the CAB. CAB is indicated in boxed region. Enlarged view of the boxed region showing more clearly the cER labelled with a DiO together with the Kif2 labelling (red). At the edge of the CAB some tubes of cER are visible (arrows). Scale bars = 10µm. n>50. See Supplementary Movie 9. “

The entire z stack is also shown as a new Supplementary Movie 9, the new Legend on page 39 reads :

“Supplementary Movie 9. Kif2 localizes to cortical endoplasmic reticulum.

Isolated cortices were prepared, fixed and labelled for cortical endoplasmic reticulum (DiO, green) and with anti Kif2 (red) at the 8-cell stage. Z stacks of confocal optical sections shows the labelling pattern of the cER (green) and Kif2 (red). Note that at the edge of the CAB some tubes of cER can be seen (one is highlighted by the arrow) which are stained homogenously with DiO while the Kif2 labelling is more punctate. Scale bar = 10µm.”

Also see new sentence added to page 17 :

“As previously described^{37,46}, ER of isolated cortices was labelled with the addition of 0.2µg/ml DiO C6(3) (Invitrogen) for 1 min. following fixation and immunolabelling.”

We agree with the reviewer that diffusion versus destruction is an important point and we looked closely at this since in the past we have measured cyclin B and cyclin A destruction in the ascidian model. However, we have found no evidence in support of Kif2 being destroyed, but do present evidence in support of diffusion. In addition, we performed a ratiometric analysis of Kif2::Tom versus Plk1::Ven in a ROI containing most of B4.1 cells before and during cell division and found that there was no significant loss of Kif2 protein versus Plk1. We did not pursue this further. Please see the added text on page 9:

“It is important to note that the presence of red Kif2::Dendra on chromosomes does not rule-out destruction of Kif2 at the CAB, but it does show that some red Kif2 protein is capable of leaving the CAB. However, since ascidian Kif2 lacks a destruction box motif it is therefore unlikely to be a substrate of the anaphase-promoting complex/cyclosome which targets protein like cyclins A and B for destruction as we showed in the ascidian^{44,45}.”

More minor points:

Figure 1B: why was the same colour used for aPKC and NN18?

Figure 1B. Same color for aPKC and NN18. A new figure has been provided with different colours.

Figure 2B could be shifted to supplemental figures.

Figure 2B has been moved to the Supp. Section. See new Supp Fig. 5.

Figure 2D: it would be nice to mention that the N-ter domain of Kif2 has previously been shown to influence its subcellular localization.

We thank the reviewer for prompting us to cite an article detailing the role of the N-terminal domain of MCAK in directing the subcellular localization of MCAK (Ems-McClung et al., 2007). This reference has been added to the text at page 8. “**The N-terminal domain of MCAK is involved in subcellular targeting⁴³ which is consistent with our findings that the N-terminal fragment is sufficient for driving CAB localization.**”

Finally, the form of the manuscript could be significantly improved. For instance, the layout of the figures could be made more professional (and more accurately referenced in the text). The text should be expurgated from very technical descriptions. To encourage potential readers, the abstract should in particular avoid excessive use of acronyms and symbols.

We have changed the figures as asked by the reviewer and have also focused on being more thorough with our citing of the figures in the text. Please see many changes in red throughout the manuscript. We have also gone through the results section to remove technical descriptions.

Reviewer #3 (Remarks to the Author):

In the manuscript by Costache and colleagues, they have used the ascidian, a classic model system for asymmetric segregation of cellular contents during development, to explore the mechanism by which cortically induced microtubule polymerization is used to promote asymmetric spindle positioning. The authors have discovered, using an antibody against the mammalian kinesin-13, Kif2A, and also via live imaging of a cloned ascidian Kif2 construct fused to fluorescent proteins, that Kif2 associates with mitotic structures expected based on the mammalian kinesin-13 literature. Significantly, Kif2 also strongly associated in a cell cycle dependent manner with the ER-rich centrosome-attracting body (CAB) which is implicated in positioning of the spindle for asymmetric cell divisions during early cleavage in ascidians. The authors hypothesize that the appearance of a member of the kinesin-13 microtubule depolymerizing family on the CAB might reflect the mechanism by which the spindle asters become asymmetric during unequal cleavage as one centrosome approaches the CAB. They have performed two experiments to bolster this claim. They have applied nocodazole to a restricted region of the cortex near the CAB to experimentally induce the movement of the spindle toward the CAB. They have also engineered a dominant-negative Kif2 construct which lacks putative MT depolymerizing activity but still associates with the CAB and have recorded that the distance between the CAB and the CAB-proximal spindle pole increased suggesting that the asymmetric spindle localizing machinery was impaired, an observation that was phenocopied by taxol administration.

I am enthusiastic about this study for a number of reasons. First, the localization of kinesin-13 members in mammalian cells have been reported on membrane-bound organelles (namely lysosomes) however their function there is not well understood because kinesin-13s lack transport

activity. The present manuscript describes a novel and compelling activity for membrane-bound kinesin-13 activity in controlling the position of microtubule structures by spatially adjusting microtubule length. The association of Kif2 with the ER-rich CAB is particularly intriguing because enzymatic activation of Kif2A has recently been reported by an Arf-GAP associated with endosomes (Luo et al. 2016 JBC). Thus, association with the CAB has the potential to also be activating for Kif2's enzymatic activity - similar to the classic idea of "cargo activation" for conventional kinesin. Second, the authors are correct in stating that asymmetric spindle positioning in other, more well-studied systems have consistently described a requirement for kinesin-13 family members and microtubule depolymerization without identifying a good molecular mechanism for spatially selective destabilization of microtubules within the spindle. It is known in systems such as *C. elegans*, that a kinesin-13 is required for asymmetric spindle positioning but whether it is generally needed or specifically required in a spatially significant way is wholly unknown. The regulated association of a kinesin-13 family member on the surface of the CAB provides us with visually arresting insight into how this occurs in a cytoplasmically specialized organism. This can provide impetus for examining this mechanism in other cells where kinesin-13 family members may be more subtly employed in selective MT depolymerization.

I would like to see a couple of questions addressed prior to publication:

We thank the reviewer for suggesting interesting experiments and for carefully reviewing our manuscript.

1-The authors applied nocodazole to the cortex opposite the proximal spindle pole and convincingly measured a decrease in distance between the pole and the CAB. Since the CAB is an area of localized MT depolymerizing activity (at least at certain times), this adds potential redundancy to the experiment. Did the authors try this experiment on non-CAB containing blastomeres as well? In other words, is an attractive force from the CAB necessary in addition to regulation of MT length? Cortical dynein contributes localized pulling forces and has been implicated in both asymmetric and symmetric spindle positioning. Thus, one would think that dynein is present and capable of pulling spindles off center in both CAB-containing blastomeres and those which divide symmetrically. Yet, the results in Figure 6Bii suggest that other blastomeres are immune to nocodazole-dependent relocalization. To understand this issue in more detail I would also like some clarification on the timing of nocodazole application versus the timing with which Kif2 dissociates from the CAB. I believe that these questions can be addressed by clarification in the text.

We have done the nocodazole needle experiment in non-CAB blastomere and the data are shown in Supplementary Figure 8. Also, see new text on page 12/13 :

“Placing nocodazole pipettes on non-CAB blastomeres did not cause spindle poles to move towards the nocodazole needle (Supplementary Fig. 8).”

Also see new Supp. Fig. 8 Legend page 36 :

“**Supplementary Figure 8.** Nocodazole pipette applied to non-CAB blastomeres (A4.1).

Embryos at the 8-cell stage (interphase) were bathed in Cell Mask orange to label the mitotic spindle poles and the plasma membrane and during mitosis a nocodazole pipette was applied to the surface of one A4.1 blastomere immediately following NEB. Note the exaggerated movement of the spindle pole nearest the nocodazole pipette away from nocodazole pipette. We measured the distance to the midline before and after application of the pipette. The

distance to the midline before pipette was $19.2 \pm 2.0 \mu\text{m}$ and increased to $22.8 \pm 1.7 \mu\text{m}$ (mean \pm s.e.m. n=6). Time in min. Scale bar = $50 \mu\text{m}$.”

Also see discussion page 14 :

“Our results have led us to the conclusion that polymerization of astral microtubules opposes the pulling forces that likely displace the mitotic spindle towards the CAB. However, it is currently unknown how the CAB pulls the spindle towards the cortex⁵⁶ although this will be a key area for future studies.”

2-Similarly, is it possible to rescue the dominant-negative effect of the expressed mutant Kif2 protein with locally applied nocodazole? If not, does this indicate that DN Kif2 eliminates both MT-depolymerizing activity and attractive pulling forces on the spindle toward the CAB? I am not expecting this experiment to be performed as a condition of publication, however, I would like to understand in more detail why this experiment is not present.

We thank the reviewer for pointing out the association of Kif2 with the endosome protein AGAP1. We have added the following sentences and reference (Luo et al., 2016) to the discussion, page 14 :

“Interestingly, it was recently found that in human cells Kif2A can also associate with organelles⁵⁵. For example, Kif2A associates with a sub-type of Arf GAPs (AGAP1) that is found on endosomes⁵⁵ and this association between Kif2A and AGAP1 is involved in cytoskeletal remodeling and cell movement.”

It is not currently known if there is a pulling force associated with the CAB and we are currently working on this issue. We have therefore added the following additional sentence to page 14 for clarity :

“Our results have led us to the conclusion that polymerization of astral microtubules opposes the pulling forces that likely displace the mitotic spindle towards the CAB. However, it is currently unknown how the CAB pulls the spindle towards the cortex⁵⁶ although this will be a key area for future studies.”

Although an excellent suggestion of rescuing the DN-Kif2 with the pipette, given the current available techniques (microinjection followed by micromanipulation in a time-constrained scenario) this experiment is beyond our technical capabilities.

In conclusion, this manuscript presents a number of technically challenging studies performed in a comparative system - a primitive chordate - that presents uniquely informative specialized features. Investigation of the activities associated with the CAB presents an opportunity for researchers, who possess the skills to utilize this system, to experimentally explore the spatial regulation of MT length - which is a key feature of many general cellular processes.

Reviewers' Comments:

Reviewer #1 (Remarks to the Author)

Review of revised manuscript by Costache et al.

The authors altered text and figures to address most of the concerns that the three reviewers raised on their original submission. The manuscript is strengthened as a result. Nevertheless, some of the points that I raised in my initial review need further attention before publication in Nature Communication can be endorsed.

Main points:

- Figure 3A: the authors added this important new panel in response to point 2 of my original review. However, they should report the ratiometric signal all the way to anaphase, and not stop at prometaphase, as this will allow readers to better assess the relationship between the loss of Kif2 and the movement of the aster towards the CAB.

- The author might want to consider that instead of "opposing the pulling forces provided by the CAB" (as stated on page 13), local microtubule depolymerization may be generating the force that pulls the aster towards the CAB. This seems like the most parsimonious explanation for their findings.

- Supplementary Figure 4: this new figure panel is not satisfactory as such. There is no DNA counterstain to assess the stage of the cell cycle of the blastomeres, which has a major impact on the microtubule array. Moreover, a Z stack should be provided so as to assess the entire microtubule cytoskeleton. In short: more convincing data is needed to assert that overexpression of Kif2 impairs the microtubule cytoskeleton.

- The experiment reported in Figure 6C should be quantified, as the authors have done in Figure 6B iv (related to point 1 of my original review).

Minor point:

- Although the authors have improved the writing in some places, the results section in particular would benefit greatly from further polishing. Several of the newly introduced sentences are poorly linked to the rest of the text (just to give an example, on page 6, the new text defines the CAB... even though it is already defined in the previous sentence). Page 9 is another example where further polishing/compacting would be beneficial.

- On page 4, the authors mention that the Dynactin/Dynein complex protein DRBY-1 interacts with Pins and NuMA at the posterior cortex of *C. elegans* embryos. This interaction is not limited to the posterior cortex, and does not necessarily involve direct interaction with DRBY-1. The statement should be corrected accordingly.

- On page 10, first line, the authors mention that Kif2 occupies the thicker cER layer, and mention Figure 2A. I guess Figure 4A is what is meant here?

- Figure 4B: higher magnification views should be provided to be able to discern the punctate staining that is said to be present.

- Page 11, towards the bottom of the page: it is unclear why low levels of H2B::mRFP1 would help distinguish the two groups of injected eggs. Please explain.

- Figure 3C, right: what is the very strong red signal close to the cortex?

- Figure 4B, typo: "reticulum"
- Figure 4B and 4C: please mention the color in which DiO can be seen.
- Supplementary Figure 6 is spectacular, but lacks a scale bar and an indication of where the CAB is located.
- Figure 2A: why is there an "hs" mention in "anti-hsKIF2A? Are the embryos not stained with antibodies raised against Phallusia Kif2?
- Figure 2B and 2C: although the authors indicate the number of embryos scored in the legend, they do not comment on whether they all conformed to the images shown in the figure panels. Please provide this information.
- Figure 3B: the mentions of PEM1 and aPKC have disappeared from the corresponding quantifications, on the right.
- In response to my comment 9 in the original review, the authors added a sentence on page 8 to mention that it is unclear why the C-terminal dimerization alone cannot target to the CAB. Although this is helpful, it appears that they have gone only half way; they should spell out explicitly that this domain is expected to form heterodimers with the endogenous protein.

Reviewer #2 (Remarks to the Author)

After first revision, this technically challenging study provides a very elegant and nicely written description of the dynamic localisation and function of ascidian Kif2 in spindle pole positioning during early unequal cell division. I am enthusiastic about the work and support its publication in Nat. Comm. The authors may choose to take the following - mostly formal - remarks into account:

- 1) It is a pity that the authors do not provide the gene model identifier (or unique gene ID) for the *Ciona* and *Phallusia* Kif2 genes. Addition of these identifiers (Ciinte.g00008837 and phmamm.g00002556?) in the methods section will make sure this nice work is properly curated and integrated into databases.
- 2) I am not sure why the authors have not gone one step beyond spindle localisation and measured the volumes of the cells after division, at least in Figure 6 A and B. Is spindle localisation a sufficient proxy for unequal cleavage?
- 3) The authors say at the bottom of page 11 that none of the DN-Kif2 injected embryos made it to the 32-cell stage. Does this mean that Kif2 has additional roles during the cell cycle? What happens to these embryos?
- 4) Which phase of the cell cycle is shown on Figure 2B?
- 5) Could the authors provide a reference for the caged Combretastatin?

Reviewer #3 (Remarks to the Author)

I am very satisfied with the revision of this manuscript. My concerns have been addressed. I think this manuscript represents an important comparative study in a classic system for unequal cell division and unequal partitioning of cellular determinants.

Nat Comm Article Costache et al.

We thank all three reviewers for a thorough analysis of the strengths and weaknesses of our article, and in particular for the many comments that have greatly strengthened the article.

Responses

Reviewer #1 (Remarks to the Author):

Review of revised manuscript by Costache et al.

The authors altered text and figures to address most of the concerns that the three reviewers raised on their original submission. The manuscript is strengthened as a result. Nevertheless, some of the points that I raised in my initial review need further attention before publication in Nature Communication can be endorsed.

Main points:

- Figure 3A: the authors added this important new panel in response to point 2 of my original review. However, they should report the ratiometric signal all the way to anaphase, and not stop at prometaphase, as this will allow readers to better assess the relationship between the loss of Kif2 and the movement of the aster towards the CAB.

Please see amended Figure 3A with extra data points added.

- The author might want to consider that instead of "opposing the pulling forces provided by the CAB" (as stated on page 13), local microtubule depolymerization may be generating the force that pulls the aster towards the CAB. This seems like the most parsimonious explanation for their findings.

We thank the reviewer for the comment. We have added the following sentence to page 15 in order to make the discussion more balanced. “Furthermore, since MCAK can produce a significant pulling force during microtubule depolymerization⁵⁸ it is also possible that some of the remaining Kif2 localized at the CAB creates a pulling force as the microtubules touching the CAB depolymerize. “

- Supplementary Figure 4: this new figure panel is not satisfactory as such. There is no DNA counterstain to assess the stage of the cell cycle of the blastomeres, which has a major impact on the microtubule array. Moreover, a Z stack should be provided so as to assess the entire microtubule cytoskeleton. In short: more convincing data is needed to assert that overexpression of Kif2 impairs the microtubule cytoskeleton.

We thank the reviewer for pointing this out. We have included a better example of Kif2-induced microtubule depolymerization. See new Supplementary Figure 4 and new Supplementary Movie 14 for both datasets: zygote overexpressing Kif2 and zygote without extra Kif2. The pronuclei are indicated showing that both zygotes are an equivalent cell cycle stage (first interphase). Supplementary Figure Legend 4 has been changed and a new Supplementary Movie 14 and accompanying Figure Legend have

been added (both in red for clarity).

- The experiment reported in Figure 6C should be quantified, as the authors have done in Figure 6B iv (related to point 1 of my original review).

We have added the quantification of these data to Figure 6 C(ii) and also the text in the Figure Legend:

Figure Legend 6 C. “(ii) Spindle-pole-cortex distance following uncaging of Combretastatin was $11.7 \pm 0.9 \mu\text{m}$ (mean \pm s.e.m, n=16) versus $15.2 \pm 0.4 \mu\text{m}$ (mean \pm s.e.m, n=32) without the Combretastatin. Student’s *t*-test, $**P < 0.005$. n=16.

Minor point:

- Although the authors have improved the writing in some places, the results section in particular would benefit greatly from further polishing. Several of the newly introduced sentences are poorly linked to the rest of the text (just to give an example, on page 6, the new text defines the CAB... even though it is already defined in the previous sentence). Page 9 is another example where further polishing/compacting would be beneficial.

We have gone through the results section again to improve the newly-inserted text. See amendments in red throughout.

- On page 4, the authors mention that the Dynactin/Dynein complex protein DRBY-1 interacts with Pins and NuMA at the posterior cortex of *C. elegans* embryos. This interaction is not limited to the posterior cortex, and does not necessarily involve direct interaction with DRBY-1. The statement should be corrected accordingly.

Please see new sentence on page 4:

“The dynein light chain protein DYRB-1 coupled to GFP has been demonstrated to co-immunoprecipitate with endogenous LIN-5 and GPR-1/2 in *C. elegans* embryos thus suggesting that DYRB-1 may provide a physical link between the endogenous dynein/dynactin complex and either LIN-5 or GPR-1/2²⁴. However, this interaction has not been shown to be limited to the posterior cortex “

- On page 10, first line, the authors mention that Kif2 occupies the thicker cER layer, and mention Figure 2A. I guess Figure 4A is what is meant here?

Please see amended sentence on page 10:

“Unlike aPKC protein which labels the cortex (Figure 1B), Kif2 protein occupies the thicker cER layer (Figures 4 A). “

- Figure 4B: higher magnification views should be provided to be able to discern the punctate staining that is said to be present.

New insets of highlighted area have been added to Figure 4B.

- Page 11, towards the bottom of the page: it is unclear why low levels of H2B::mRFP1 would help distinguish the two groups of injected eggs. Please explain.

New sentence has been added to page 12 :

“ In order to distinguish those eggs injected with a mixture of wild type Kif2 plus Ens::3GFP from those injected with DN-Kif2 plus Ens::3GFP (since both batches will display green fluorescence), we mixed low levels of H2B::mRFP1 mRNA with DN-Kif2/Ens mRNA before microinjection (DN-Kif2 eggs thus also display red fluorescence). “

- Figure 3C, right: what is the very strong red signal close to the cortex?

Strong signal in Figure 3 C is the CAB – the figure and figure legend have been annotated in order to make this clearer.

- Figure 4B, typo: "reticulum"

Changed to reticulum.

- Figure 4B and 4C: please mention the color in which DiO can be seen.

Color has been stated. See figure legend 4B:

“Schematic of cortical preparation, ER in red. **Top right:** Probing cortical preparations with **DiO (green) to label the endoplasmic reticulum and Kif2 antibody (red) revealed** that Kif2 protein was localized to the domain of cortical ER in the CAB. CAB is indicated in boxed region. **Lower:** Enlarged views of the boxed region **in top right images** showing more clearly the cER labelled with a DiO (**green**) together with the Kif2 labelling (red). At the edge of the CAB some tubes of cER are visible (**insets of small boxed regions**) where **Kif2 fluorescence appears punctate relative to the green DiO cER signal.** “

- Supplementary Figure 6 is spectacular, but lacks a scale bar and an indication of where the CAB is located.

20µm scale bar has been added to Supp Fig. 6. See Figure Legend : “Scale bar = 20 µm”.

(The CAB is not indicated because it is not present in an unfertilized eggs where the cortical ER forms a thin layer around most of the egg cortex. The CAB forms following fertilization due to an actomyosin-driven cortical contraction that causes this thin layer of cortical ER, shown in the figure, to accumulate near the vegetal pole.)

- Figure 2A: why is there an "hs" mention in "anti-hsKIF2A? Are the embryos not stained with antibodies raised against Phallusia Kif2?

Please see information in the materials and methods section. Human antibodies are purified with Phallusia Kif2 protein. Page 17:

“...anti Kif2 (1/200 following affinity purification of anti-human Kif2⁶⁵ on *Phallusia* Kif2 protein produced in bacteria),

- Figure 2B and 2C: although the authors indicate the number of embryos scored in the legend, they do not comment on whether they all conformed to the images shown in the figure panels. Please provide this information.

Figure legend 2B and 2C (and 2A) have been amended to contain the necessary information:

“Images are representative of all embryos”

- Figure 3B: the mentions of PEM1 and aPKC have disappeared from the corresponding quantifications, on the right.

This was an error when formatting (we thank the reviewer for noting this error): PEM1 and aPKC have been added back to the Figure. Please see Figure 3B.

- In response to my comment 9 in the original review, the authors added a sentence on page 8 to mention that it is unclear why the C-terminal dimerization alone cannot target to the CAB. Although this is helpful, it appears that they have gone only half way; they should spell out explicitly that this domain is expected to form heterodimers with the endogenous protein.

A new sentence has been added to page 8:

“However, since the C-terminal coiled-coil domain of MCAK drives dimerization⁴³, it is not clear why the C-terminal domain of ascidian Kif2 does not localize to the CAB by forming a dimer with endogenous Kif2 in the CAB. “

Reviewer #2 (Remarks to the Author):

After first revision, this technically challenging study provides a very elegant and nicely written description of the dynamic localisation and function of ascidian Kif2 in spindle pole positioning during early unequal cell division. I am enthusiastic about the work and support its publication in Nat. Comm. The authors may choose to take the following - mostly formal - remarks into account:

1) It is a pity that the authors do not provide the gene model identifier (or unique gene ID) for the *Ciona* and *Phallusia* Kif2 genes. Addition of these identifiers (Ciinte.g00008837 and phmamm.g00002556?) in the methods section will make sure this nice work is properly curated and integrated into databases.

This was an oversight. Both unique gene IDs have been added to the materials and methods section:

“All Kif2 constructs were prepared using *Phallusia mammillata* Kif2 (unique gene ID: phmamm.g00002556) and *Ciona intestinalis* Kif2 (unique gene ID: Ciinte.g00008837). “

And to the results section where Kif2 is first mentioned:

“There is only one member of the Kif2 family in the ascidian (*P. mammillata*: PmKif2 : **unique gene ID: phmamm.g00002556** and *C. intestinalis*: CiKif2 : **unique gene ID: Ciinte.g00008837**) and other non-vertebrate deuterostomes (Supplementary Fig. 2),...”

2) I am not sure why the authors have not gone one step beyond spindle localisation and measured the volumes of the cells after division, at least in Figure 6 A and B. Is spindle localisation a sufficient proxy for unequal cleavage?

This is something we would like to do – however, because the blastomeres are misshapen, even during mitosis (see our recent article in eLife Dumollard et al., 2017), measuring volumes accurately would require precise segmentation of datasets containing many more z-planes than we have (here we favored temporal over spatial measurements). For this reason we have measured only spindle positioning. In answer to the second point, in a previous article we showed that the cleavage furrow forms around the midpoint of the spindle – see Prodon et al., 2010, Figure 6Aii: we therefore believe that spindle position is a good proxy for cleavage furrow position.

3) The authors say at the bottom of page 11 that none of the DN-Kif2 injected embryos made it to the 32-cell stage. Does this mean that Kif2 has additional roles during the cell cycle? What happens to these embryos?

We think that that once over a threshold level DN-Kif2 affects all blastomeres by causing excessive microtubule accumulation (much like taxol), although we did not pursue these later effects.

4) Which phase of the cell cycle is shown on Figure 2B?

We thank the reviewer for pointing this out: the full movie of Figure 2B (8-cell stage) can be seen in Supplementary movie 7 (with bright-field overlay also included for clarity) and this has been clarified in the Figure legend: the image in Figure 2B is prometaphase. The other two images in Figure 2B show the interphase localization pattern. Figure legend 2B has been amended as follows :

“Live *Phallusia* embryos expressing Kif2::Tom (red) and the microtubule markers EB3::GFP (green, 8-cell stage **prometaphase, also see Supplementary Figure 7 where bright-field data has been included**) or Ens::3GFP (green, 16-cell stage and 32-cell stage **interphase**)...”

5) Could the authors provide a reference for the caged Combretastatin?

A reference to caged combretastatin has been added to page 13 and to the reference list :

“To depolymerize microtubules via a second method we used photo-activation of caged Combretastatin⁴⁷...”

Also see reference list :

47. Wühr, M., Tan, E. S., Parker, S. K., Detrich, H. W. & Mitchison, T. J. A model for cleavage plane determination in early amphibian and fish embryos. *Curr. Biol. CB* **20**, 2040–2045 (2010).

Reviewer #3 (Remarks to the Author):

I am very satisfied with the revision of this manuscript. My concerns have been addressed. I think this manuscript represents an important comparative study in a classic system for unequal cell division and unequal partitioning of cellular determinants.